# MQuAKE-Remastered: Multi-Hop Knowledge Editing Can Only Be Advanced with Reliable Evaluations

**Shaochen (Henry) Zhong**[*][♣], **Yifan Lu**[*][♣], **Lize Shao** [♣], **Bhargav Bhushanam** [∞], **Xiaocong Du** [∞], **Yixin Wan** [†], **Yucheng Shi** [◇], **Daochen Zha** [♣], **Yiwei Wang** [†], **Ninghao Liu** [◇], **Kaixiong Zhou** [♡], **Shuai Xu** [♠], **Kai-Wei Chang** [†], **Louis Feng** [∞], **Vipin Chaudhary** [♠], and **Xia Hu** [♣]

♣ Department of Computer Science, Rice University
◇ School of Computing, University of Georgia
♡ Department of Electrical and Computer Engineering, North Carolina State University
♠ Department of Computer and Data Sciences, Case Western Reserve University
† Department of Computer Science, University of California, Los Angeles
∞ Meta Platforms, Inc.

## Abstract

Large language models (LLMs) can give out erroneous answers to factually rooted questions either as a result of undesired training outcomes or simply because the world has moved on after a certain knowledge cutoff date. Under such scenarios, *knowledge editing* often comes to the rescue by delivering efficient patches for such erroneous answers without significantly altering the rest, where many editing methods have seen reasonable success when the editing targets are simple and direct (e.g., *"what club does Lionel Messi currently play for?"*). However, knowledge fragments like this are often deeply intertwined in the real world, making effectively propagating the editing effect to non-directly related questions a practical challenge (to entertain an extreme example: *"What car did the wife of the owner of the club that Messi currently plays for used to get to school in the 80s?"*[1]). Prior arts have coined this task as *multi-hop knowledge editing* with the most popular dataset being MQuAKE, serving as the sole evaluation benchmark for many later proposed editing methods due to the expensive nature of constructing knowledge editing datasets at scale. In this work, we reveal that **up to 33% or 76% of MQuAKE's questions and ground truth labels are, in fact, corrupted in various fashions due to some unintentional clerical or procedural oversights**. Our work provides a detailed audit of MQuAKE's error pattern and a comprehensive fix without sacrificing its dataset capacity. Additionally, we benchmarked almost all proposed MQuAKE-evaluated editing methods on our post-fix dataset, **MQuAKE-Remastered**. We observe that many methods try to overfit the original MQuAKE by exploiting some dataset idiosyncrasies of MQuAKE. We provide a guideline on how to approach such datasets faithfully and show that a simple, minimally invasive approach — **GWalk** — can offer beyond SOTA editing performance without such exploitation. The MQuAKE-Remastered datasets and utilities are available at `huggingface.co/datasets/henryzhongsc/MQuAKE-Remastered` and `github.com/henryzhongsc/MQuAKE-Remastered`, respectively.

## 1 Introduction

Given the widespread public-facing popularity of various Large Language Model-powered (LLM) products (Zhao et al., 2023; Yang et al., 2024b), even an occasional user has likely experienced LLMs giving out erroneous answers to factually rooted, knowledge-intensive questions. While why LLMs would hallucinate such kind of misinformation is complex and still an open problem —

---

* Equal contribution. Work corresponds to Shaochen (Henry) Zhong <shaochen.zhong@rice.edu> and done while in collaboration with Meta.

[1] For interested readers, it was a Rolls-Royce: `youtube.com/watch?v=DbwiHC1Fu-E&t=132s`.

noisy training data, model bias, out-of-distribution questions, or even simply because the world has moved on after a certain knowledge cutoff date... all likely contributed their fair share to this rather undesired character of LLMs (Huang et al., 2023; Zhang et al., 2023) — **under a practical context,** *knowledge editing* **is often considered the go-to remedy by delivering efficient patches for such erroneous answers** without significantly altering the LLM's output on unrelated queries, nor undergoing another extensive pretraining or finetuning section (Sinitsin et al., 2020; Mitchell et al., 2022).

With the growing need for more credible and trustworthy LLMs, a vast amount of LLM-specific knowledge editing methods have been proposed, and many of them have seen reasonable success in addressing simple and direct editing targets. For example, most modern knowledge editing methods can reliably edit the answer of *"What club does Lionel Messi currently play for?"* from *"Paris Saint-Germain"* to *"Inter Miami CF"* and therefore correctly reflecting the occupation status of Messi (Zhong et al., 2023).

## 1.1 MULTI-HOP KNOWLEDGE EDITING POSES PRACTICAL SIGNIFICANCE AND NON-TRIAL CHALLENGES.

However, due to the intertwined nature of different knowledge fragments, a small change in one knowledge fragment can produce ripple-like effects on a vast number of related questions (Zhong et al., 2023; Cohen et al., 2023). It is often a non-trivial challenge to efficiently propagate the editing effect to non-directly related questions with proper precision and locality. E.g., — as an intentionally extreme case — *"What car did the wife of the owner of the club that Messi currently plays for used to get to school in the 80s?"* Many knowledge-edited LLMs can still struggle while being fully aware of Messi's abovementioned club transfer (Zhong et al., 2023).

Prior arts have realized the practical significance of being able to edit such complex/non-direct questions upon a certain knowledge update, as different knowledge fragments are almost always deeply entangled with each other in the real world (Zhong et al., 2023; Cohen et al., 2023; Wei et al., 2024). Meanwhile, exhausting all potential combinations of questions related to one or a few updated knowledge fragments is impractical. Even if it is feasible, this poses high operational costs, and a repeated effort would be required should Messi ever opt to transfer again.

Intuitively, a practical knowledge editing method needs to produce correct answers to relevant factual questions with only a few updated knowledge fragments available. This task has been coined as *multi-hop knowledge editing* or *MHKE*, **with the founding, most popular, and only publicly available reflective dataset to date being MQUAKE by Zhong et al. (2023); serving as the sole evaluation backbone for many proposed modern editing methods** due to the expensive nature of making counterfactual and temporal datasets at such a scale (>10,000 cases provided, see Table 7).

## 1.2 UNFORTUNATELY, MQUAKE IS FLAWED DUE TO UNINTENTIONAL CLERICAL AND PROCEDURAL ERRORS — WE FIXED/REMADE IT AND RE-BENCHMARKED ALMOST ALL PROPOSED MULTI-HOP KNOWLEDGE EDITING METHODS.

While MQUAKE is the founding dataset of multi-hop knowledge editing tasks and very much brings life to this vital subject, through a comprehensive audit, we reveal that **up to 33% or 76% of MQUAKE questions and ground truth labels are, in fact, corrupted in various fashions due to some unintentional clerical or procedural errors**; which inevitably cast doubts on the effectiveness of the developed methods evaluated on MQUAKE. The issues with MQUAKE are significant and growing, especially as MQUAKE becomes a widely adopted (and often solely relied on) evaluation in the editing community. Given its importance for building more reliable LLMs — a critical aspect of NLP development — we present our work to advance multi-hop knowledge editing with the following contributions:

- **A comprehensive audit of MQUAKE:** We are the first to present a comprehensive audit of the existing errors within MQUAKE (Zhong et al., 2023), bringing awareness to the knowledge editing community regarding this popular dataset with significant task importance attached.
- **Fix/remake MQUAKE to MQUAKE-Remastered:** We present the only available fix/remake that not only patches all discovered errors, and done so without sacrificing the intended intensity and capacity of the original MQUAKE whenever possible.
- **Extensively re-benchmark of almost all existing multi-hop knowledge editing methods:** Given the currently existing reports based upon the original MQUAKE are flawed reflections

of such proposed methods' capability, we additionally re-benchmark almost all existing multi-hop knowledge editing methods that are available against our MQUAKE-REMASTERED datasets.

- **Present a faithful yet beyond SOTA pilot method for future multi-hop knowledge editing development.** We observe that many proposed MHKE methods intentionally or unintentionally overfit the original MQUAKE dataset by applying data-specific operations that are largely unique to the MQUAKE dataset family. We provide guidance on how to approach these datasets faithfully and additionally show that a simple, minimally invasive method — GWalk — with no such idiosyncrasies-overfitting operations can also achieve beyond-SOTA editing performance.

## 2 PRELIMINARY

### 2.1 BACKGROUND OF MQUAKE

MQUAKE (Multi-hop Question Answering for Knowledge Editing) is a knowledge editing dataset focusing on the abovementioned multi-hop question answering tasks proposed in Zhong et al. (2023), where every case of MQUAKE is a multi-hop question made by a chain of single-hop subquestions. Specifically, MQUAKE is constructed based on the Wikidata:RDF dataset (Vrandečić & Krötzsch, 2014), which, in its rawest format, is a knowledge graph consisting 15+ trillion of Resource Description Framework (RDF) triples. MQUAKE essentially builds a much more concise subgraph with only 37 manually elected common relations and top 20% of the most common entities, where a walk of $\{2, 3, 4\}$-hop on this subgraph can form a case (which is a chain of $\{2, 3, 4\}$ single-hop subquestions connected together) in the MQUAKE dataset.

MQUAKE is presented as two sub-datasets: MQUAKE-CF and MQUAKE-T. The former focuses on counterfactual tasks, while the latter on temporal changes. We highlight that there is also a MQUAKE-CF-3K dataset, a subset of MQUAKE-CF that only contains 3,000 cases out of the original 9171 cases. Authors of MQUAKE evaluate their proposed method, MeLLo (Zhong et al., 2023), upon this MQUAKE-CF-3K dataset, which then becomes an unspoken standard for the later proposed multi-hop knowledge editing methods (Gu et al., 2024; Shi et al., 2024; Wang et al., 2024; Chen et al., 2024; Cheng et al., 2024). Due to the popularity of this sub-sampled dataset, we provide our error analysis mostly based on MQUAKE-CF-3K and MQUAKE-T in the following §3. For interested readers, we additionally provide the same error analysis upon the full MQUAKE-CF in the Appendix C.3. We also collect the dataset statistics in Table 7.

### 2.2 EVALUATING USING MQUAKE

Datasets like MQUAKE-CF and MQUAKE-CF-3K are often tested under varying "editing intensities" based on the number of cases considered "edited." This simulates different levels of deviation between the model's learned knowledge and the newly edited information. This approach is effective because strong knowledge editing methods should handle both large-scale updates and smaller, more localized edits, ensuring that the changes do not interfere with unrelated knowledge.

In its original paper, MQUAKE-CF-3K is evaluated when $\{1, 100, 1000, 3000\}$ of its 3,000 cases are edited, similarly, MQUAKE-T is evaluated when $\{1, 100, 500, 1868\}$ of its 1,868 cases being edited, forming an experiment report like Table 6. This kind of report granularity is also adopted by the majority of later proposed multi-hop knowledge editing methods, either in full (Chen et al., 2024) or in spirit with different "$k$-edited" settings (Gu et al., 2024; Wang et al., 2024; Shi et al., 2024; Cheng et al., 2024; Mengqi et al., 2024). In this work, we report at an even finer level of "$k$-edited" granularity for maximum cross-reference potentials.

## 3 AUDITING MQUAKE

In this section, we present a comprehensive audit of the error pattern that existed in MQUAKE-CF-3K and MQUAKE-T (Zhong et al., 2023). We specifically note that our audit is there to provide a better understanding to the knowledge editing community, especially when digesting methods evaluated on these datasets. **Our audit is not to discredit the contribution of MQUAKE, or any of the proposed methods evaluated on MQUAKE.** We recognize that no dataset can be perfect, especially when it is intrinsically hard to collect large-scale counterfactual and temporal datasets.

### 3.1 INTRA CONTAMINATION BETWEEN EDITED CASES AND UNEDITED CASES

As discussed in §2.2, having a gradual evaluation coverage from a few to all cases being edited like Table 6 makes sense as an evaluation granularity. However, one critical issue is that $k \in \{1, 100, 1000, 3000\}$-edited cases (supposed MQUAKE-CF-3K) are randomly sub-sampled from

the 3,000 total cases. Thus, **there is no guarantee that the $k$-edited cases and $(3000 - k)$ unedited cases would require one consistent set of knowledge and, therefore, risk contamination.**

Figure 1: Example of contamination between an edited case and an unedited case

For a concrete example, consider the above two multi-hop questions from MQUAKE-CF-3K illustrated in Figure 1. When case 482 is selected as an edited case, the edited fact in case 482 (*"Kamal Hassan is a citizen of the USA"*) will contaminate the unedited case 126 since both multi-hop questions involve a subquestion inquiring along the knowledge of "The citizenship of Kamal Haasan", where the corresponding edited fact would be retrieved. This leads to the model generating an answer in conflict with MQUAKE-CF-3K's label, causing inaccurate experiment readings. See Appendix C.1 for a detailed walk-through. We note the above-illustrated contamination is not a cherry-picked fluke, but rather a wild-spread error. Here, we sample $\{1, 100, 1000, 2000, 3000\}$-editing targets from MQUAKE-CF-3K using random seed `100`, and summarize the following error statistics in Table 1.

Table 1: Error statistics of MQUAKE-CF-3K and MQUAKE-T (Zhong et al., 2023) in terms edited cases contaminating unedited cases. $k$-edited means $k$ cases out of the total dataset are edited.

| # of Contaminated | MQUAKE-CF-3K | | | | | MQUAKE-T | | | |
|---|---|---|---|---|---|---|---|---|---|
| | 1-edit | 100-edit | 1000-edit | 2000-edit | 3000-edit | 1-edit | 100-edit | 500-edit | 1868-edit |
| Cases | 0 | 2,013 | 1,772 | 910 | 0 | 29 | 1421 | 1327 | 0 |
| Subquestions | 0 | 2,706 | 3,075 | 1,664 | 0 | 29 | 1421 | 1327 | 0 |

It is observable from Table 1 that **even a small number of edited cases will cause concerningly large contamination to unedited cases and subquestions, where 67% and 76% of all cases from MQUAKE-CF-3K and MQUAKE-T are contaminated with just 100 cases being edited**, introducing a significant distortion to the reported experiment results conducted in this fashion.

We also note that this contamination decreases as the number of edited cases ($k$-edit) increases, but it's simply a result of fewer unedited cases being available for contamination as $k$ grows. For example, in the extreme case of 3000-edit, there is no contamination between edited and unedited cases in MQUAKE-CF-3K because all cases are edited. However, 3000-edit has the highest level of contamination within edited cases, which we explore further in §3.2.

One additional note-worthy detail is in the original MQUAKE literature (Zhong et al., 2023) and most works following its setup, "$k$-edit" (or "editing batch size" in some literature) means only $k$ of edited cases are evaluated, without any unedited cases tested. Thus, by such evaluation design, there is no Intra Contamination as discussed in this section since no edited cases are evaluated with any unedited case involved. In this work, we evaluated both types of cases to better reflect the locality of different knowledge editing methods (i.e., even in the "1-edit" setting of MQUAKE-CF-3K or MQUAKE-REMASTERED-CF-3K, we evaluate the 1 edited cases as well as the 2,999 unedited cases). **We consider evaluating an editing method's performance with both unedited and edited cases while having edited facts present a paramount setting.** Since only edited cases

are being evaluated, one might easily end up with a method that is proficient at dealing with edited facts at the cost of the inability to answer questions citing unedited knowledge. This is practically catastrophic under a real-world scenario as typically there are way more unedited facts - e.g., Earth is mostly round[2] — than edited facts. Only evaluating on edited cases might present a false sense of reliability. In fact, we found that existing knowledge editing methods often fail by retrieving unrelated edited facts when a subquestion demanding unedited knowledge is inquired. See plots highlighting "Incorrect Unedited Retrieval" in Appendix H.

## 3.2 INNER CONTAMINATION BETWEEN DIFFERENT EDITED CASES

Contamination might also happen among multiple edited cases because a certain subquestion presented in different edited cases can be edited in some but unedited in others[3] as illustrated in Figure 2. Similar to §3.1, an edited fact from case 1968 would alter the answer to an unedited hop in the edited case 1570. So, the model would generate an answer in conflict with the dataset ground truth label, causing inaccurate experiment readings. See Appendix C.2 for a detailed walk-through.

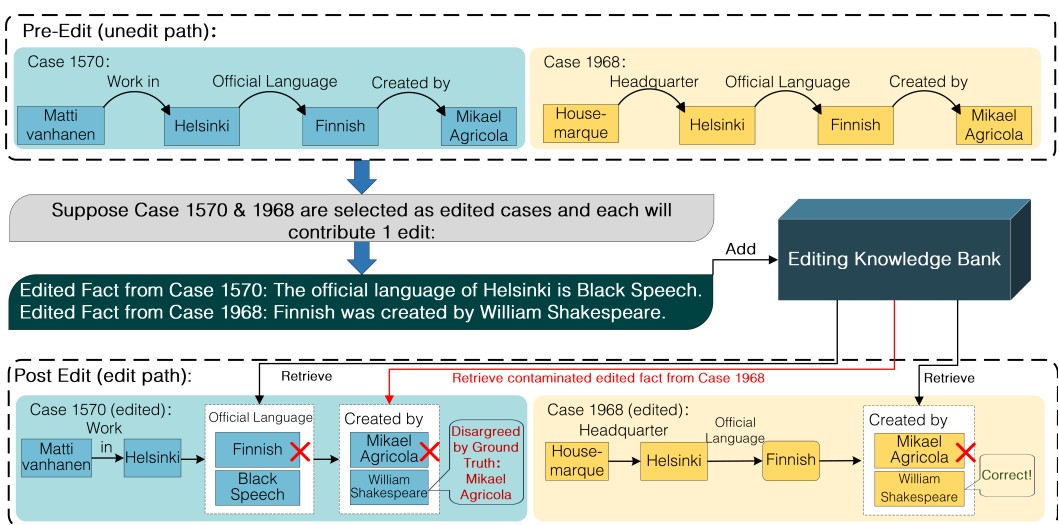

Figure 2: Example of contamination between two edited cases

This type of contamination is, once again, universally visible in MQUAKE as shown in Table 2; which is very much a flipped version of Table 1. With $k$-edit growing, there are more edited cases, thus more edited-to-edited contamination. Notably, **under the 3000-edit tasks, almost one-third (998/3000, $\approx$33%) of the evaluated cases are contaminated**, which again introduces distortion to the reported experiment results. We omit the report on MQUAKE-T here because there is only one edit-to-edit contamination when all 1,868 cases from MQUAKE-T are edited (`case_id:424`).

Table 2: Error statistics of MQUAKE-CF-3K (Zhong et al., 2023) in terms edited cases contaminating each others. $k$-edited means $k$ cases out of the total 3,000 cases are edited.

| # of Contaminated | 1-edit | 100-edit | 1000-edit | 2000-edit | 3000-edit |
|---|---|---|---|---|---|
| Cases | 0 | 14 | 265 | 619 | 998 |
| Subquestions | 0 | 14 | 337 | 854 | 1,399 |

## 3.3 CONFLICTING EDITS

The two types of contamination introduced in §3.1 and §3.2 are indeed subtle and hard to detect. However, MQUAKE-CF-3K also includes some straightforward edit conflicts, such as for the subquestion *"Which company is Ford Mustang produced by?"* we have the following edits:

⬦ `case_id:2566` (edited): ~~Ford Moter Company~~ Nintendo.
⬦ `case_id:231/2707` (edited): ~~Ford Moter Company~~ Fiat S.p.A.

This is going to cause a direct conflict when `case_id:2566` and any of the `case_id:231` or `case_id:2707` are both selected as edited cases, as they shall confuse any knowledge edited

---

[2]We can't speak for the future, but let's hope this fact stays unedited.

[3]Note, an edited case does not require all of its subquestions being edited, but merely one of it (Table 7).

LLM for having two different answers to the same questions. Fortunately, such types of errors are rather minuscule in MQUAKE-CF-3K, with the abovementioned Ford Mustang question and three cases being the only affected data samples.

### 3.4 MISSING INFORMATION IN MULTI-HOP QUESTION INSTRUCTIONS

As mentioned in §2, the MQUAKE dataset is built upon a severely filtered Wikidata:RDF knowledge graph (Vrandečić & Krötzsch, 2014). Specifically, the triples of a certain $\{2, 3, 4\}$-hop walk on this subgraph are then fed into a `GPT-3.5-turbo` model to generate three multi-hop question instructions in a natural language format. During evaluation, an LLM is considered right should it correctly answer against any one of the three multi-hop question instructions (Zhong et al., 2023).

However, while repeating generation three times definitely reduces the chances of having incomprehensible question instructions, we noticed some of such instructions in MQUAKE are still incomplete. We take `case_id:546`'s triple set and its generated 3 question instructions as an example:

- `case_id:546` (unedited): We have a 2-hop question with "Albert Mohler" as the subject and `(employer, religion or worldview)` as the relation chain. MQUAKE-CF-3K provides the following generated multi-hop questions:
  ◇ *Generation #1: What religion is Albert Mohler associated with?*
  ◇ *Generation #2: Which religion does Albert Mohler follow?*
  ◇ *Generation #3: With which religious faith does Albert Mohler identify?*

All three generated questions omit the part mentioning which company/institution Albert Mohler is employed by and essentially reduce themselves to single-hop questions, where a correct generation should read like *"What religion is Albert Mohler's employer associated with?"* and it is not equivalent to any of the above three instructions. Without the complete question instruction, suppose there is an edit on Albert Mohler's employer (which there indeed are some), the final answer would likely change. However, with this omission of information, even the best knowledge-edited LLM cannot answer the question correctly with a faithful approach because Alber's employer is not in the picture.

As a general analysis, we find **the natural language question instructions of 672 cases in MQUAKE-CF-3K are missing information in comparison to their raw triplet chain.** This number is counted in the sense that one or more pieces of information present in the triple chain are missing from all three variants of the generated natural language instruction. Similarly, there are 2,830 and 233 cases of erroneous instructions in MQUAKE-CF and MQUAKE-T, respectively.

### 3.5 DUPLICATED CASES

The last kind of error we discovered in MQUAKE is simply unintended duplication — i.e., two or more cases sharing the same start subject, edited facts, chain of triples, and the final answer — i.e., they are the carbon copy of each other, yet simultaneously exist in the dataset. We discovered **47, 4, and 4** cases of duplication, respectively, in MQUAKE-CF, MQUAKE-CF-3K, and MQUAKE-T.

## 4 REMASTERING MQUAKE

In this section, we illustrate how we modified and improved the MQUAKE dataset to MQUAKE-REMASTERED with various fixes on the data samples themselves, as well as providing utility modules to facilitate how one interacts with such datasets. We provide an audit correctness analysis in Appendix D. Furthermore, we demonstrate the impact of our improvements through ablation studies that analyze the types of errors addressed, as discussed in Appendix E.

### 4.1 HARD CORRECTIONS

Three types of error existing in MQUAKE can be fixed once and for all with some careful hard corrections, they are namely Conflicting Edits (§3.3), Missing Information in Multi-hop Question Instructions (§3.4), and Duplicated Cases (§3.5). For Conflicting Edits and Duplicated Cases, since there are only a few such errors ($<50$ per type per dataset), we employ some manual corrections to address them: in the former case, we flip the minority edits to align with the majority edits (and adjust their answers to their subsequence subquestions, should there be any); in the latter case, we simply remove such duplicated cases (for MQUAKE-CF-3K, we then manually select 4 more cases from MQUAKE-CF to keep the dataset having 3,000 cases in total and a 1,000 cases for $\{2, 3, 4\}$-hops — as the original MQUAKE-CF-3K intended). For errors regarding Missing Information in Multi-hop Question Instructions, we rewrite the natural language question instructions for all cases in the datasets using meta-llama/Llama-3.1-405B (Dubey et al., 2024) for consistency sake.

### 4.2 DYNAMIC MASKING FOR MAXIMUM COVERAGE: MQUAKE-REMASTERED-CF, MQUAKE-REMASTERED-CF-3K, AND MQUAKE-REMASTERED-T

Due to the contamination count of Intra Edited-to-Unedited Contamination (§3.1) and Inner Edited-to-Edited Contamination (§3.2) tend to grow in the opposite direction as shown in Table 1 and 2, it is impossible to find a fix within the current MQUAKE that can address both issues without significantly decreasing the dataset size. As an alternative, we develop an API that will take a `case_id` and an `edited_flag` as input, indicating the evaluating case-in-question and whether this case is considered edited; our API shall then return a set of triples that are contamination free by dynamically masking out the conflicting edits from other cases. After such, the user may build up an editing knowledge bank upon such returned triplets and conduct evaluations for any memory-based knowledge editing methods without losing any cases caused by contamination. Due to the nature of the $N$-hop question, at most $N$ edited facts would be removed for each case, a marginal reduction compared to (typically thousands of) total edited facts as summarized in Table 13.

Specifically, once `case_id`-of-interest is given, our API would loop through all of its subquestions and identify if any is considered edited under another case. If there is a hit, the triple for such edited subquestions is removed from the bank of edited triples. This dynamic masking mechanism would ensure all cases within the original MQUAKE be usable against memory-based knowledge editing methods. **However, the drawback of masking is it won't support parameter-based knowledge editing methods**, where weight update is required. We additionally provide a MQUAKE-REMASTERED-CF-6334 to address the need for such methods (Appendix F.1).

## 5 MAKING SAFE AND FAITHFUL APPROACH TO MQUAKE AND MQUAKE-REMASTERED

In addition to our benchmark results we'd show in §6, it is our observation that many MHKE methods with decent performance on MQUAKE or MQUAKE-REMASTERED are (often coincidentally) utilizing designs that leverage dataset idiosyncrasies unique to MQUAKE. For example, methods like RAE (Shi et al., 2024) and GLAME (Mengqi et al., 2024) utilize Wikidata (Vrandečić & Krötzsch, 2014) as the external KG to better detect the edit-induced conflicts, which happen to be the source of MQUAKE as discussed in §2.1 — where whether sourcing the same Wikidata KG as MQUAKE might bring them data-specific advantages remains to be confirmed. Similarly, PokeMQA (Gu et al., 2024) utilizes the 6,218 cases included in MQUAKE-CF but not in MQUAKE-CF-3K as the train set to train its auxiliary components, then tested on MQUAKE-CF-3K. Given MQUAKE is a dataset with relatively low diversity (e.g., with only 37 types of relations), whether having a heavily overlapped train and test set will result in data-specific advantages unique to MQUAKE and its variants, again remains to be confirmed (we also note, without considerations like MQUAKE-REMASTERED-CF-6334, this training split is surely contaminated across and within the two sets, which would likely introduce conflicting knowledge via its training scheme).

On the same note, important consideration when using a KG to store unedited facts — a common approach in methods like RAE (Shi et al., 2024), GLAME (Mengqi et al., 2024), and StruEdit (Bi et al., 2024) — is the additional operational cost of maintaining a large unedited KG. Since unedited facts typically far outnumber edited ones, managing such a KG requires continuously reconciling it with the KG storing edited facts. That said, one could fairly argue that storing unedited and edited facts allows for the gradual expansion of a unified, ever-evolving "ground truth for all" KG, eliminating the need to maintain separate model-specific edited KGs for each LLM. However, effectively querying such a vast unified KG and adapting its knowledge to LLMs presents additional challenges, particularly if the method also aims to leverage an LLM's internal knowledge. We highlight this issue because there are clear trade-offs between these two approaches, and they are rarely discussed in the existing MHKE literature. While we chose not to use an external unedited KG in our pilot method, we believe it is both fair and beneficial to acknowledge this consideration.

### 5.1 A MINIMALLY INVASIVE BUT PERFORMANT APPROACH: **GWALK**

Here, we provide a brief walkthrough of a simple method we designed, namely GraphWalk. GWalk does not leverage any idiosyncrasies unique to MQUAKE or MQUAKE-REMASTERED. Yet, it still presents beyond SOTA performance, surpassing many, if not all, established baselines. **We illustrate this pilot method as concrete guidance and potential inspiration to our future multi-hop knowledge editing scholars.** The design of GWalk hinges on the fundamental pipeline of memory-based knowledge editing methods: where the pool of source only contains *edited facts*.

This school of editing methods has proven to be successful, mainly because it can leverage the power of retrieval-argument generation (RAG) combined with the in-context learning (ICL) capability of LLMs, yet an edited-only knowledge bank is typically minuscule, much like done so in MeLLo (Zhong et al., 2023). Different from MeLLo, where all edited facts are converted from triples to natural language (NL) descriptions in its edited bank, **GWalk preserves the edited facts in their original triples fashion and leverages the graph topology.** This makes maintaining this edited bank much easier — as one can easily adjust the entity or relation on a knowledge graph without rewriting every natural language description of every related edited fact. **It also brings more precise retrieval mapping when a question of a certain edited fact is asked.** Methods like MeLLo (Zhong et al., 2023) rely on RAG from a pool of edited facts in NL format. This can lead to unintended retrievals, where irrelevant facts with similar embeddings are retrieved, potentially causing hallucinations. However, if we simply query the entity and relations implied in a question against a knowledge graph, there is less chance of retrieving unintended materials. We share the detailed pseudocode of GWalk in Algorithm 1 and demonstrate some further studies in Appendix H.

---

**Algorithm 1:** General Procedure of GWalk on One Multi-hop Question

---

**Input:**
    $M$, the Question Answering Language Model;
    $T$, a Text-embedding model;
    $Q$, a Multi-hop Question;
    $E$, a bank of edited facts as a knowledge graph.
**Output:**
    $o_p$, the answer to $Q$.
**Initialize:**
    $i = 1$, the subquestion counter;
    $o_p$ = None, the answer from the previous subquestion.

1   $s \leftarrow$ Extracted subject from $Q$;
2   $rels \leftarrow$ Prompt $M$ to breakdown $Q$ into a sequence of relations.
    `/* If` $Q$ `is 'What is the official language of the country where Karl Alvarez`
    `holds citizenship?', then` $s$ `would be 'Karl Alvarez' and a possible` $rels$ `is`
    `['citizenship', 'official language']`                                  `*/`
3   **for** $r \in rels$ **do**
4      Query $< s, r, ? >$ against $E$ using $T$, namely we do $T(s)$ first to determine if there is a retrievable $s \in E$, then inspect if the $s \in E$ has an relation edge retrievable by $T(r)$.
        `/* We set a threshold on embedding similarity for` $T$ `to determine whether an`
        `item is retrievable or not.`                                `*/`
5      Prompt $M$ to generate subquestion $q_i$ with $s$ and $r$.
6      $o_p \leftarrow$ the $M$-generated answer to $q_i$.
7      **if** $T(s, r)$ has a valid retrieval $< s, r, o^* >$ **then**
8          $o_p \leftarrow o^*$;
        `/* The answer to this subquestion will be the start subject of the next`
        `subquestion.`                                      `*/`
9      $s \leftarrow o_p$ ;
10     $i \leftarrow i + 1$;
11   **Return** $o_p$;

---

## 6   Benchmark and Discussion

Given almost all proposed MHKE methods are (often solely) evaluated on the original, error-contained, MQUAKE datasets. Here, we provide a re-benchmark of those methods against our post-fix MQUAKE-REMASTERED datasets to provide some more reliable results.

**Compared Methods** In this work, **we aim to cover most, if not all, open-sourced knowledge editing methods specifically evaluated on the original MQUAKE.** This includes MeLLo (Zhong et al., 2023), PokeMQA (Gu et al., 2024), RAE (Shi et al., 2024), and DeepEdit (Wang et al., 2024) as methods specifically proposed to target this MHKE problem and evaluated on MQUAKE. We additionally include ICE (Cohen et al., 2023) and IKE (Zheng et al., 2023a) as these are also methods purposed for the (single-edit) MHKE task, though not specifically evaluated on MQUAKE in their original publications. General editing methods like ROME (Meng et al., 2022) and MEND (Mitchell et al., 2022) are also featured. We note that we are aware methods like GLAME (Mengqi et al., 2024), StableKE (Wei et al., 2024), Temple-MQA (Cheng et al., 2024), GMeLLo (Chen et al., 2024), and StruEdit (Bi et al., 2024) are also evaluated on MQUAKE, but they are purposely omitted from our re-benchmark coverage due to lack of timely open-sourced implementation, likely because many of these works are still in-submission by our work's first appearance.

**Covered Models** We opt to use lmsys/vicuna-7b-v1.5 (Zheng et al., 2023b), mistralai/Mistral-7B-Instruct-v0.2 (Jiang et al., 2023), and meta-llama/Meta-Llama-3-8B-Instruct (AI@Meta, 2024) as the choice of question-answering models, both for alignment with existing works (Zhong et al., 2023; Shi et al., 2024; Gu et al., 2024) as well as providing coverage the most recent language

models. For methods that require a text-embedding model as a retriever, we use facebook/contriever-msmarco (Izacard et al., 2022) for alignment with MeLLo (Zhong et al., 2023).

**Covered Datasets** We will provide coverage on our post-fix dataset, namely MQUAKE-REMASTERED-CF, MQUAKE-REMASTERED-CF-3K, and MQUAKE-REMASTERED-T in the masking fashion illustrated in §4.2; as well as MQUAKE-REMASTERED-CF-6334 in its vanilla form. These datasets are respectively corresponding to the original MQUAKE-CF, MQUAKE-CF-3K, and MQUAKE-T from Zhong et al. (2023) (with 6334 as an extra for parameter-based methods), but with the types of error mentioned in §3 fixed in the via means illustrated in §4.

## 6.1 RESULTS AND DISCUSSION

Table 3: Performance Comparison of Original MQUAKE and our MQUAKE-REMASTERED datasets using vicuna-7b-v1.5 (Zheng et al., 2023b). For the CF-3k dataset families, we utilized the 1000-edit setting, whereas for the T dataset families, we applied the all-edit setting. The original MQUAKE cannot faithfully reflect the true capacities of the methods due to errors specified in §3, especially if the method-in-question is performant.

| Method | MQuAKE-CF-3k | | | MQuAKE-T | |
| | Original | Original-v2 | Remastered | Original | Remastered |
|---|---|---|---|---|---|
| MeLLo (Zhong et al., 2023) | 14.4 | 14.1 | **14.63** | 30.84 | **44.37** |
| GWalk (Ours) | 36.07 | 41.0 | **57.37** | 46.41 | **54.88** |

Given that our MQUAKE-REMASTERED is mostly provided as a fix to MQUAKE, we would like to highlight the drastic result differences when the same method is evaluated on these two datasets (Table 3); where the differences are especially pronounced for stronger methods. Such differences suggest all previous reporting on MQUAKE has room for reliability improvements, which we filled here with MQUAKE-REMASTERED.

Table 4: Experiments on MQUAKE-REMASTERED-CF-6334 with numbers of edited cases and methods.
*Total Accuracy*
Results are reported in the format: *(Test Edited Accuracy, Train Edited Accuracy, Unedited Accuracy)*

| Method | MQUAKE-REMASTERED-CF-6334 | | | |
| | 100-edit | 1000-edit | 3000-edit | 6344-edit |
|---|---|---|---|---|
| *lmsys/vicuna-7b-v1.5 (Zheng et al., 2023b)* | | | | |
| MeLLo (Zhong et al., 2023) | 19.16 (0, 10.99, 19.37) | 19.27 (5.1, 9.58, 24.53) | 11.17 (4.31, 8.55, 23.3) | 6.83 (4.58, 7.72, 19.05) |
| ICE (Cohen et al., 2023) | OOM | OOM | OOM | OOM |
| IKE (Zheng et al., 2023a) | OOM | OOM | OOM | OOM |
| PokeMQA (Gu et al., 2024) | - | - | - | 21.77 (3.25, 30.82, 1.59) |
| DeepEdit Wang et al. (2024) | <1 | <1 | <1 | <1 |
| GWalk (Ours) | **57.55** (22.22, 64.84, 57.48) | **61.79** (29.08, 66.17, 63.23) | **59.1** (39.3, 63.74, 64.33) | **56.62** (44.64, 62.11, 68.25) |
| *mistralai/Mistral-7B-Instruct-v0.2 (Jiang et al., 2023)* | | | | |
| MeLLo (Zhong et al., 2023) | 27.5 (<1, 23.08, 27.65) | 27.54 (12.76, 24, 30.4) | 24.37 (11.88, 25.51, 32.06) | 21.26 (13.29, 24.9, 30.16) |
| ICE (Cohen et al., 2023) | OOM | OOM | OOM | OOM |
| IKE (Zheng et al., 2023a) | 8.82 (11.11, 6.59, 8.86) | OOM | OOM | OOM |
| PokeMQA (Gu et al., 2024) | - | - | - | 20.38 (3.99, 27.41, 69.84) |
| DeepEdit Wang et al. (2024) | <1 | <1 | <1 | <1 |
| GWalk (Ours) | **56.25** (33.33, 57.14, 56.28) | **58.9** (34.69, 60.57, 60.6) | **56.03** (42.69, 59.04, 59.85) | **54.43** (47.49, 57.74, 52.38) |

In Table 4, we present benchmark results on MQUAKE-REMASTERED-CF-6334. GWalk consistently outperforms other methods in terms of models and edit numbers. The "OOM"s in ICE and IKE are due to memory overload from concatenating all edited facts in the in-context learning prompt. Whereas the "<1" results likely stem from the LLM's failure to recognize the few-shot examples, often generating irrelevant tokens or failing to follow the few-shot format. This issue was observed with MeLLo using Meta-Llama-3-8B-Instruct, and with DeepEdit using vicuna-7b-v1.5 and Mistral-7B-Instruct-v0.2. **Due to page limitation, we refer our readers to Appendix I for benchmarks of MQUAKE-REMASTERED-CF, MQUAKE-REMASTERED-CF-3K, and MQUAKE-REMASTERED-T.** We present MQUAKE-REMASTERED-CF-6334 in the main text solely because it can feature the most methods. Last, we highlight that due to the feedback of the reviewers and the community, as well as company policies, we have then updated our datasets for better polishness, which might mean slightly different results than currently reported as we cannot complete a full rerun before camera-ready. We will make the official run soon and update with more comprehensive experiment reports with one-click reproducible scripts in our repository.

## 7 RELATED WORKS

**Audit and Fix of MQUAKE**   To the best of our knowledge, no work has conducted a comprehensive audit to MQUAKE as we do, but two prior arts have touched on the errors existing in MQUAKE: GMeLLo (Chen et al., 2024) and DeepEdit (Wang et al., 2024).

Table 5: Comparison of error analysis/quantification/fix of MQUAKE provided in different works.

| Ref. | Error Types Found | Error Quantified | Error Scopes Fixed | Cost of Fixing |
|---|---|---|---|---|
| GMeLLo (Chen et al., 2024) | Missing Instruction | No | No | N/A |
| DeepEdit (Wang et al., 2024) | Inner Contamination | CF-3K in **3000**-edit | CF-3K in **3000**-edit | 998 out of 3000 cases removed from CF-3K |
| Ours | Intra Contamination, Inner Contamination, Conflicting Edits, Missing Instructions, Duplicated Cases | CF-3K in {**1, 100, 1000, 3000**}-edit, T in {**1, 100, 500, all**}-edit, CF-9K in {**1, 100, 1000, 3000, 6000, all**}-edit | CF-3K, T, CF-9K in **any**-edit; Remastered-CF-6334 in **any**-edit | No case removed from CF-3K, CF-9K, or T. |

Specifically, GMeLLo (specifically, its anonymous submission to ACL ARR 2024 Feb) briefly discusses the inconsistency between the triple chain and the corresponding generated instructions in its §4.5.1, which is the same type of error we discussed in §3.4. However, GMeLLo merely presents two examples of such an error without providing any quantitative error analysis or fix; we did both in §3.4 and §4.1. DeepEdit (Wang et al., 2024) discovered the same inner contamination error (edited-to-edited) as we discussed in §3.2, but limited to one dataset (MQUAKE-CF-3K) under one setting (when all 3000 cases are considered edited). Further, DeepEdit removed all 998 inner contaminated cases from the MQUAKE-CF-3K dataset — which is (supposedly) the same 998 cases we detect in Table 2 under the 3000-edit column — and named it MQUAKE-2002. While this fix is, of course, helpful, we argue **our REMASTERED fixes are much more comprehensive and effective since they patched many more errors revealed in §3 (the other four types of errors still exist in MQUAKE-2002), and most importantly, done so without scarifying almost 1/3 of the capacity of the original dataset** thanks to masking utility we proposed in §4.2. We further demonstrate the quantifiable difference between our work, GMeLLo, and DeepEdit in Table 5.

**MQUAKE-CF-3K-V2**   Other than the above two works, we have privately consulted with the first authors of MQUAKE (Zhong et al., 2023) and confirmed the legitimacy of our findings and those of the prior arts. The MQUAKE team has then rolled out their update: MQUAKE-CF-3K-V2. Upon investigation, MQUAKE-CF-3K-V2 indeed resolves the edited-to-edited Inner Contamination (§3.2) and Conflicting Edits (§3.3). However, we respectfully note it still falls short in terms of Intra Contamination as reflected in Table 11 (53%+ cases contaminated under 1000-edit), which makes it ineligible in evaluating edited and unedited cases at the same time — a paramount setting for reasons highlighted at the end of §3.1. Further, MQUAKE-CF-3K-V2 still involves unfixed Duplicated Cases (§3.5) and Missing Instructions (§3.4); more on this in Appendix C.4. Beyond unfixed errors, we note the original MQUAKE-CF-3K was subsampled from MQUAKE-CF to have 1,000 cases per {2, 3, 4}-hop in order to provide a balanced representation. However, this is no longer true under MQUAKE-CF-3K-V2 as shown in Table 7. And, as the MQUAKE-CF-3K-V2 name implied, it is error resolution is limited within the MQUAKE-CF-3K dataset, but not MQUAKE-CF nor MQUAKE-T. While we authors, again, pay our utmost respect to the MQUAKE team for bringing the MHKE task to the community, we earnestly argue MQUAKE-CF-3K-V2 offers nothing more than our MQUAKE-REMASTERED-CF-3K, as the latter fixed substantially more errors, supports an important setting, and is faithful to the balanced 1,000 cases per {2, 3, 4}-hop design.

**Benchmark and Guidance**   Our work re-benchmarks nearly all open-sourced knowledge editing methods on MQUAKE and guides on safely and faithfully approaching such datasets. To the best of our knowledge, no other work offers this level of benchmarking or touches on the same issues. Notably, we are likely the only work to evaluate on MQUAKE-CF/MQUAKE-CF-9K, the largest dataset that even the original MQUAKE paper did not assess due to resource constraints. Table 14 illustrates the significant difference in evaluation coverage between our work and previous efforts.

## 8 CONCLUSION

We audited MQUAKE and present MQUAKE-REMASTERED with many errors fixed. We re-benchmarked many MHKE methods with MQUAKE-REMASTERED and presented a simple, efficient, yet extremely capable baseline — GWalk — for future MHKE studies.

ACKNOWLEDGMENTS

This research was partially supported by NSF Awards ITE-2429680, IIS-2310260, CNS-2431516, OAC-2112606, and OAC-2117439. Furthermore, this work was supported by the US Department of Transportation (USDOT) Tier-1 University Transportation Center (UTC) Transportation Cybersecurity Center for Advanced Research and Education (CYBER-CARE) grant #69A3552348332.

Further, this work made use of the High Performance Computing Resource in the Core Facility for Advanced Research Computing at Case Western Reserve University (CWRU). We give special thanks to the CWRU HPC team for their prompt and professional help and maintenance. The views and conclusions in this paper are those of the authors and do not represent the views of any funding or supporting agencies.

Outside funding agencies, our work benefits from the great support of the knowledge editing community of LLMs. We appreciate the efforts of the PokeMQA team (Gu et al., 2024), the RAE team (Shi et al., 2024), and the DeepEdit team (Wang et al., 2024) for confirming our findings and providing support in testing their knowledge editing methods on our post-fix MQUAKE-REMASTERED datasets. Last, we give our special shout-out to **Zexuan Zhong** and **Zhengxuan Wu** — the first authors of the original MQUAKE paper (Zhong et al., 2023) — for validating our findings and sharing their insights into our work. We pay our greatest respect to them for their founding efforts in bringing the important MHKE task to the field, while being engaged and supportive of future community scrutiny and improvements.

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

# A    LIMITATIONS

While we strive to make a solid and comprehensive contribution to the MHKE field and the knowledge editing community at large, we must acknowledge certain shortcomings in our post-fix MQUAKE-REMASTERED datasets. Notably, MQUAKE-REMASTERED remains focused solely on knowledge-intensive tasks where both unedited and edited facts can be represented as triples in a KG. However, this assumption does not always hold, as some facts are not practically feasible to deconstruct into a triple.

For example, consider the question-answer pair: "Which actor in a Marvel movie has the same initial for both their first and last name?" — "Paul Rudd (for Ant-Man)." A real-world KG is unlikely to contain such a specific and elaborate relation edge. **As a result, it remains an open question of how MHKE methods would perform when faced with a multi-hop question (or a subquestion) that references such non-triple-representable unedited or edited knowledge;** or, on an even broader scale, a multifaceted question that can't be easily decomposed into multiple standalone subquestions.

This challenge is akin to the common struggle of deciding when to RAG and when to engage in CoT reasoning. Should retrieval/reasoning be applied to every query, or should there be an oversight mechanism that determines when an extra operation (in our case, an editing resolution/confirmation) should be triggered, as explored in works like Self-RAG (Asai et al., 2023)? Right now, MHKE methods typically check for editing resolutions for every subquestion (and assume such a subquestion breakdown is obtainable in the first place), which is a mechanism with strong assumptions and requires significant support if it is to be deployed at scale.

Although this limitation is not unique to MQUAKE-REMASTERED but rather a general shortcoming of almost all works in MHKE, we believe it is both fair and necessary to highlight it, as it warrants dedicated studies of its own. We are encouraged to see the community moving in a more comprehensive direction, with evaluations like DUnE (Akyürek et al., 2023) being constructed. We expect a reasonable complement to simply evaluate MHKE methods — with edited facts present — on typical datasets that demand no editing operations and observe if they significantly degrade an LLM's performance.

## B    EXTENDED PRELIMINARY

### B.1    DEMO REPORT OF MQUAKE

Table 6: Standard reporting format of MQUAKE-CF-3K, and MQUAKE-T demoed with MeLLo on Vicuna-7B (Zheng et al., 2023b); $k$-edited means $k$ cases out of the total cases are edited. Abbreviated table courtesy of Zhong et al. (2023) (Table 3).

| Model | Method | MQUAKE-CF-3K | | | | MQUAKE-T | | | |
|---|---|---|---|---|---|---|---|---|---|
| | | 1-edit | 100-edit | 1000-edit | 3000-edit | 1-edit | 100-edit | 500-edit | 1868-edit |
| Vicuna-7B | MeLLo (Zhong et al., 2023) | 20.3 | 11.9 | 11.0 | 10.2 | 84.4 | 56.3 | 52.6 | 51.3 |

### B.2    DATASET STATISTICS

Table 7: Dataset Statistics of MQUAKE. Numbers are in terms of cases (a case in MQUAKE is a chain consisting of multiple subquestions).

| Dataset | # of Edits | 2-hop | 3-hop | 4-hop | Total |
|---|---|---|---|---|---|
| MQUAKE-CF-3K | 1 | 513 | 356 | 224 | 1,093 |
| | 2 | 487 | 334 | 246 | 1,067 |
| | 3 | - | 310 | 262 | 572 |
| | 4 | - | - | 268 | 268 |
| | All | 1,000 | 1,000 | 1,000 | 3,000 |
| MQUAKE-CF-3K-V2 | 1 | 599 | 423 | 51 | 1,073 |
| | 2 | 536 | 374 | 136 | 1,046 |
| | 3 | - | 339 | 229 | 568 |
| | 4 | - | - | 313 | 313 |
| | All | 1,135 | 1,136 | 729 | 3,000 |
| MQUAKE-CF | 1 | 2,454 | 855 | 446 | 3,755 |
| | 2 | 2,425 | 853 | 467 | 3,745 |
| | 3 | - | 827 | 455 | 1,282 |
| | 4 | - | - | 436 | 436 |
| | All | 4,879 | 2,535 | 1,804 | 9,218 |
| MQUAKE-T | 1 (All) | 1,421 | 445 | 2 | 1,868 |

Table 8: Dataset Statistics of MQUAKE-REMASTERED. Numbers are in terms of cases (a case in MQUAKE is a chain consisting of multiple subquestions).

| Dataset | # of Edits | 2-hop | 3-hop | 4-hop | Total |
|---|---|---|---|---|---|
| MQUAKE-REMASTERED-CF-3K | 1 | 513 | 356 | 224 | 1,093 |
| | 2 | 487 | 334 | 246 | 1,067 |
| | 3 | - | 310 | 262 | 572 |
| | 4 | - | - | 268 | 268 |
| | All | 1,000 | 1,000 | 1,000 | 3,000 |
| MQUAKE-REMASTERED-CF | 1 | 2,446 | 850 | 441 | 3,737 |
| | 2 | 2,415 | 852 | 463 | 3,730 |
| | 3 | - | 823 | 451 | 1,274 |
| | 4 | - | - | 430 | 430 |
| | All | 4,861 | 2,525 | 1,785 | 9,171 |
| MQUAKE-REMASTERED-T | 1 (All) | 1,421 | 441 | 2 | 1,868 |
| MQUAKE-REMASTERED-CF-6334 | 1 | 1,971 | 77 | 0 | 2,048 |
| | 2 | 2,415 | 476 | 14 | 2,905 |
| | 3 | - | 823 | 128 | 951 |
| | 4 | - | - | 430 | 430 |
| | All | 4,386 | 1,376 | 572 | 6,334 |

## C    EXTENDED AUDITING

### C.1    EXAMPLE OF INTRA CONTAMINATION BETWEEN AN EDITED CASE TO AN UNEDITED CASE (§3.1)

For a concrete example, consider the following two multi-hop questions from MQUAKE-CF-3K (we also additionally provide the subquestion breakdown and intermediate answers of the two questions for better presentation, we note that such auxiliary information is not part of the instruction visible to the question-answering LLM):

- `case_id:126` (unedited): *What is the continent of the country where Kamal Haasan holds citizenship?*
    - ◇ What is the country of citizenship of Kamal Haasan? India.
    - ◇ What is the continent of India? Asia.

- `case_id:482` (unedited): *What is the capital of the country where Kamal Haasan holds citizenship?*
    - ◇ What is the country of citizenship of Kamal Haasan? India.
    - ◇ What is capital of India? New Delhi.

The correct pre-edited answer should be *"Asia"* and *"New Delhi"* respectively. As Kamal Haasan is an Indian citizen, India is located in Asia and is the capital of New Delhi. However, suppose `case_id:482` is sampled as an edited case while `case_id:126` remains unedited, we will be provided with the additional triple containing the knowledge of *"The official language of United States of America is Arabic."*

Since the unedited `case_id:126` and the edited `case_id:482` share the same subquestion of *"What is the country of citizenship of Kamal Haasan?"* The answer of `case_id:482` will be rightfully updated to *"USA"* per the new knowledge. However, the unedited `case_id:126` still considers the original answer *"India"* to be correct, and is therefore contaminated by the edited case `case_id:482` in an unintended fashion. This is problematic because a successful knowledge editing method should be able to retrieve the edited knowledge — *"Kamal Haasan is a citizen of USA?"* — upon the relevant questions (in this case the shared one), and thus answering *"North America"* to `case_id:126`. This is technically correct, but in conflict with MQUAKE-CF-3K's label, causing inaccurate experiment readings.

### C.2    EXAMPLE OF INNER CONTAMINATION BETWEEN DIFFERENT EDITED CASES (§3.2)

Again, we walk through two cases from MQUAKE-CF-3K as a concrete example. First, we show them in their unedited format (again, subquestion breakdowns and intermediate answers are here for demonstration purposes and are not visible to the question-answering LLM during evaluation):

- `case_id:1570` (unedited): *Who was the creator of the official language used in the work location of Matti Vanhanen?*
    - ◇ Which city did Matti Vanhanen work in? Helsinki.
    - ◇ What is the official language of Helsinki? Finnish.
    - ◇ Who was Finnish created by? Mikael Agricola.

- `case_id:1968` (unedited): *Who created the official language of Housemarque's headquarters location?*
    - ◇ Which city is the headquarter of Housemarque located in? Helsinki.
    - ◇ What is the official language of Helsinki? Finnish.
    - ◇ Who was Finnish created by? Mikael Agricola.

Suppose `case_id:1570` and `case_id:1968` are both selected as editing cases, two triples containing the following knowledge will be available: *"The official language of Helsinki is Black Speech"* (intended for `case_id:1570`), and *"Finnish was created by William Shakespeare"* (intended for `case_id:case_id:1968`), leading to the following edited breakdown.

- `case_id:1570` (edited): *Who was the creator of the official language used in the work location of Matti Vanhanen?*
    - ◇ Which city did Matti Vanhanen work in? Helsinki.

- ◇ What is the official language of Helsinki? ~~Finnish~~ Black Speech.
  - ◇ Who was ~~Finnish~~ Black Speech created by? J. R. R. Tolkien.
- `case_id:1968` (edited): *Who created the official language of Housemarque's headquarters location?*
  - ◇ Which city is the headquarter of Housemarque located in? Helsinki.
  - ◇ What is the official language of Helsinki? Finnish.
  - ◇ Who was Finnish created by? ~~Mikael Agricola~~ William Shakespeare.

Much like the previous conflict between unedited and edited cases, these two edited cases share a common subquestion: *"What is the official language of Helsinki?"* However, such subquestion is edited in `case_id:1570` while unedited in `case_id:1968`, causing unintended contamination.

## C.3 ERROR ANALYSIS OF MQuAKE-CF

Table 9: Error statistics of MQuAKE-CF (Zhong et al., 2023) in terms of edited cases contaminating unedited cases §3.1. $k$-edited means $k$ cases are edited out of the total 9218 cases.

| # of Contaminated | MQuAKE-CF | | | | | | |
|---|---|---|---|---|---|---|---|
| | 1-edit | 100-edit | 1000-edit | 2000-edit | 3000-edit | 5000-edit | 9218-edit |
| Cases | 62 | 3307 | 5275 | 5110 | 4578 | 3346 | 0 |
| Subquestions | 62 | 4525 | 8751 | 8989 | 8326 | 6364 | 0 |

Table 10: Error statistics of MQuAKE-CF (Zhong et al., 2023) in terms edited cases contaminating each others §3.2. $k$-edited means $k$ cases are edited out of the total 9218 cases.

| # of Contaminated | 1-edit | 100-edit | 1000-edit | 2000-edit | 3000-edit | 5000-edit | 9218-edit |
|---|---|---|---|---|---|---|---|
| Cases | 0 | 8 | 192 | 441 | 732 | 1397 | 2873 |
| Subquestions | 0 | 12 | 270 | 606 | 1027 | 1986 | 4250 |

## C.4 ERROR ANALYSIS OF MQuAKE-CF-3K-V2

MQuAKE-CF-3K-V2 (Zhong et al., 2023) is a dataset based on MQuAKE-CF-3K aimed to fix the dataset corruption issues we identified in §3. It fixed the inner-contamination (§3.2) and conflict edit errors (§3.3). However, as shown below, this dataset is still vulnerable to intra-contamination as illustrated in §3.1. Meanwhile, MQuAKE-CF-3K-V2 has **17** duplicate cases (§3.5) and **633** cases of missing information in the problem instructions (§3.4).

Table 11: Error statistics of MQuAKE-CF-3K-V2 (Zhong et al., 2023) in terms of edited cases contaminating unedited cases §3.1. $k$-edited means $k$ cases are edited out of the total 3000 cases.

| # of Contaminated | MQuAKE-CF-3K-V2 | | | | |
|---|---|---|---|---|---|
| | 1-edit | 100-edit | 1000-edit | 1000-edit | 3000-edit |
| Cases | 0 | 1643 | 1591 | 830 | 0 |
| Subquestions | 0 | 2028 | 2398 | 1357 | 0 |

## D   ERROR DETECTION PROCEDURE AND POST-AUDIT CHECKING

In this section, we discuss how exactly we carry out our audit and fixes and how we conduct our post-audit checking to ensure our audited datasets are error-free to the best of our ability.

### D.1   INTRA AND INNER CONTAMINATION

As discussed in §3.1 and §3.2, we observed that some edited facts were retrieved for subquestions that were not intended to involve an edit. We categorized this issue as contamination, where edited facts inadvertently influence the correct reasoning path. To carry out the audit, we made the following observation: regardless of whether a case is edited or unedited, a valid reasoning path must always exist from the initial subquestion to the last subquestion. Thus, suppose any unedited subquestion on this reasoning path shares the same subject and relation with a triple reflecting an edited fact; then this unedited subquestion is contaminated and therefore flagged.

We programmed the abovementioned filtering mechanism and identified the contaminated edit facts against different subquestions/cases. We then employed the API described in §4.2 to dynamically mask out contaminated cases. Last, we confirmed that there is no contamination remaining by re-executing our filtering program upon the dynamically masked dataset.

### D.2   CONFLICTING EDITS

As illustrated in §3.3, we noticed some edits within the editing knowledge bank are self-contradicting, where edits with the same subject and relation led to different tail entities. Again, we follow the intended reasoning path as introduced above and check if there are multiple edit-reflecting triples that share the source and relation with an edited subquestion. If so, this suggests there are conflicting edits. We flagged all those edited triples, put ones with shared sources and relations into the same group, then flipped the minority edit to the majority edit and updated their subquestions accordingly. We then reran the program to ensure no more flagged triples.

### D.3   MULTI-HOP QUESTION INSTRUCTION REWRITE

As highlighted in §3.4, we identified some questions lacked a complete set of relations in their instructions, thus essentially omitting necessary information for a model to provide the correct answer. We collect a list of synonyms of all relations of an editing path, then evaluate if a certain instruction is not using any of the corresponding synonyms when its reasoning path indicates it should reflect a certain relation.

Subsequently, we prompted the original meta-llama/Llama-3.1-405B to regenerate all instructions with the few-shot demonstration prompt demoed in Appendix F.2 and reran the detection procedure. This process resulted in only a small number of instructions that still didn't meet our predefined rules due to the fact that our lists of synonyms per each relation cannot be exhaustive by design. We then manually inspect and, in a few occasions, manually fix those flagged cases.

### D.4   DUPLICATE CASES

Upon investigating conflicting edits, we accidentally discovered that there exist cases with identical reasoning paths to each other, as illustrated in §3.5. We simply opt to retain only one of such cases and remove the duplicated rests. We then keep track of a set of reasoning paths from all cases and see if the cardinality of the set is equivalent to the number of cases.

# E ERROR TYPE ABLATION STUDY

In this section, we provide ablation studies demonstrating the benefits of addressing errors in the MQuAKE dataset, aligning with the observed error patterns in Tables 2 and 1. Using our proposed GWalk and the Llama-3.1-8B-Instruct model, we evaluate datasets corrected for major error types, including Inner Contamination, Intra Contamination, and Missing Information in Multi-hop Question Instructions. These error types significantly affect the performance of both edited and unedited accuracies. We opt to exclude minor errors, such as duplicate questions and conflict edits, which are automatically addressed across all settings due to their limited prevalence in the original datasets.

The observed impact is consistent with our analysis: The Inner Contamination fix has the most impact when editing intensity is high (e.g., 3000-edit). Yet, the Intra Contamination fix has the most impact with lower editing intensity (e.g., 100-edit). The Missing Instruction fix consistently improves performance across all editing intensities.

Table 12: Performance comparison across dataset variants of MQUAKE-CF-3K on meta-llama/Llama-3.1-8B-Instruct. Results are reported as $_{(Test\ Edited\ Accuracy,\ Unedited\ Accuracy)}^{Total\ Accuracy}$ .

| Type of Errors Fixed | MQUAKE-CF-3K | | |
|---|---|---|---|
| | 100-edit | 1000-edit | 3000-edit |
| Meta-Llama/Llama-3.1-8B-Instruct (Dubey et al., 2024) | | | |
| None | 45.47 (38, 45.72) | 42.73 (41.3, 43.45) | 39.57 (39.57, -) |
| Inner Contamination | 46.83 (70, 46.03) | 53.3 (73.9, 43) | 71.36 (71.36, -) |
| Intra Contamination | 71.17 (37, 72.35) | 61.73 (40.9, 72.15) | 39.87 (39.87, -) |
| Missing Instruction | 49.33 (41, 49.62) | 47.2 (45.6, 48) | 45.1 (45.1, -) |
| All (Our proposed) | **76.83** (69, 77.1) | **75.03** (74.6, 75.25) | **71.53** (71.53, -) |

# F EXTENDED REMASTERING

## F.1 CONTAMINATION FREE SUBSET: MQUAKE-REMASTERED-CF-6334

While MQuAKE-REMASTERED-MASKED with masking operation can well support memory-based knowledge editing methods, it will not be compatible with parameter-based methods. This is because, for parameter-based methods, the set of edited facts used for training and evaluation needs to be constant yet consistent with each other at all times; whereas dynamic masking cannot suffice as it is essentially adjusting the dataset on the fly during inference time.

To effectively evaluate parameter-based knowledge editing methods, we present MQUAKE-REMASTERED-CF-6334. MQUAKE-REMASTERED-CF-6334 is a dataset extracted from MQUAKE-CF, where all 6,334 cases are edited cases; and they are completely contamination-free from each other. This dataset is suitable for LLM editing with parameter-based approaches, as one can make careful splits among the 6,334 cases of MQUAKE-REMASTERED-CF-6334 to serve as train, validation, and evaluation sets.

Table 13: The number of unique edited facts for a varied number of edited cases in MQUAKE-REMASTERED-CF

| Number of Edited Cases | 100 | 1000 | 3000 | 6000 | All (9171) |
|---|---|---|---|---|---|
| **Number of Unique Edited Facts** | 150 | 1171 | 2991 | 5137 | 7252 |

Table 14: Experiment coverage comparison among our and other works. For brevity and better relevance, "Method Coverage" only includes open-sourced methods specifically designed for multi-hop editing, as adopted single-hop editors are often too weak to deliver usable results. "Separate Metrics?" means that both the accuracy of edited cases and unedited cases are reported. We consider the inclusion of both metrics paramount, as editing is often a double-edged sword, causing potential hallucinations under unedited scenarios. Prior work often only tests on the former but ignores the latter. We did both in our work.

| Ref. | Dataset Coverage | Method Coverage | Separate Metrics? | Error Fix? |
|---|---|---|---|---|
| MQuAKE (Zhong et al., 2023) | CF-3K {1, 100, 1000, all}-edit; T {1, 100, 500, all}-edit | MeLLo | No | No |
| Temple-MQA (Cheng et al., 2024) | CF-3K {1, 100, all}-edit; T {1, all}-edit | MeLLo, PokeMQA | No | No |
| Ju et al. (2024) | CF-3K {all}-edit | N/A | No | No |
| PoleMQA (Gu et al., 2024) | CF-3K {1, 100, all}-edit; T {1, 100, all}-edit | MeLLo, PokeMQA | No | No |
| Ours | CF-3K {1, 100, 1000, all}-edit; T {1, 100, 500, all}-edit; CF-9K {1, 1000, 3000, 6000, all}-edit; CF-6334 {100, 1000, 3000, all}-edit | MeLLo, ICE, IKE, PokeMQA, GWalk, RAE, DeepEdit | Yes | Yes |

## F.2 PROMPT FOR REWRITING INSTRUCTIONS

---

**Few-shot Prompt**

**Instruction:** Given a chain of relations, generate 3 multi-hop questions that comprehensively include the semantics of the relations.

**Example 1:**
*Relation Chain:*
```
XXX -> 'The author of  is' -> ' is a citizen of' -> ?
```
*Generated Questions:*

1. What is the country of citizenship of the author of XXX?

2. What country is the author of XXX a citizen of?

3. What is the nationality of the author of XXX?

**Example 2:**
*Relation Chain:*
```
XXX -> ' was developed by' -> 'The chairperson of  is' -> '
is a citizen of' -> ' is located in the continent of' -> ?
```
*Generated Questions:*

1. What continent is the country located in, where the chairperson of the developer of XXX is a citizen?

2. On which continent is the country located, whose citizen is the chairperson of the company that developed XXX?

3. Which continent houses the country of the chairperson of the developer of XXX?

**Example 3:**
*Relation Chain:*
```
<The relational chain we want the generated questions to be
based on>
```

---

# G  RIPPLEEDIT

We consider MQuAKE's task design and setup to be more reflective of real-world editing tasks, as naturally, there will always be more than one edited fact stored for any system with reasonable complexity. That being said, we are happy to report our proposed pilot method, GWalk, performs decently on RippleEdit. Here are some snapshot results on Llama-2-7b-chat:

Table 15: Single-edit result of RippleEdit-Popular/Recent/Random. C1/2 means the edit is happening at the 1st or the 2nd hop (RippleEdit cases only have 2 hops).

| Method | Popular C1 Acc. | Popular C2 Acc. | Recent C1 Acc. | Recent C2 Acc. | Random C1 Acc. | Random C2 Acc. |
|---|---|---|---|---|---|---|
| ROME | 37.4 | 16.2 | 47.8 | 50.0 | 35.5 | 49.5 |
| ICE | 85.1 | 67.6 | 74.8 | 85.0 | 73.8 | 80.3 |
| MeLLo | 45.1 | 77.1 | 50.2 | 80.0 | 40.2 | 68.3 |
| GWalk (ours) | **85.7** | **81.8** | **80.9** | **87.6** | **76.1** | **82.9** |

We additionally convert RippleEdit to a multi-edit setup — i.e., there are multiple edited facts within the editing knowledge bank at the same time — to a) make it more challenging and, b) show that our audit can also "fix" issues within a different dataset. Note we put the "fix" in quotes as RippleEdit is not designed with multi-edit in mind, so the things we fixed are not necessarily errors but just some adjustments required for making a proper multi-edit dataset. In any case, here are the snapshot results on Llama-3-8b-Instruct:

Table 16: Multi-edit result of RippleEdit-Popular/Recent/Random. In this case, we fixed 21/2/0 conflict edits and 3/120/1 case-to-case contamination within RippleEdit-Ropular/Recent/Random datasets, respectively.

| Method | Popular C1 Acc. | Popular C2 Acc. | Recent C1 Acc. | Recent C2 Acc. | Random C1 Acc. | Random C2 Acc. |
|---|---|---|---|---|---|---|
| MeLLo | 35.1 | 40.3 | 41.1 | 42.4 | 49.5 | 50.0 |
| GWalk (ours) | **79.0** | **66.9** | **79.2** | **63.9** | **72.9** | **60.0** |

## H   CASE STUDY OF GWALK

We believe GWalk is performant and practical because of two ingredients:

- It only stores edited facts in its Editing Knowledge Bank (Figure 2), contrary to some baselines (e.g., RAE (Shi et al., 2024)), where unedited facts are also stored. This is more practical to maintain as there are always fewer edited facts to keep track of, yet the total search space is much smaller, allowing more precise and efficient retrieval.
- Unlike most baselines, which store edited facts in natural language (NL) format (e.g., MeLLo (Zhong et al., 2023) and the majority of existing works) and conduct retrieval based on NL sentence embeddings, we store such editing facts on a Knowledge Graph (KG). The topology-based retrieval greatly reduces unintended retrieval, which almost always causes hallucinations.

Here is a concrete example from MQuAKE-Remastered-CF (case #16), where MeLLo retrieves an incorrect edited fact on an edited subquestion.

---

**Edited Subquestion Example**

**Question:** What is the country of citizenship of Twitter's CEO?
**1st Subquestion:** Who is Twitter's CEO?
**Generated Answer (by LLM):** Twitter's CEO is Elon Musk.
**MeLLo-retrieved edited fact:** The chief executive officer of CBS Corporation is Steve Jobs.
*// Incorrect edited fact retrieved because this edited fact is close to the subquestion from an embedding standpoint, even if it doesn't provide relevant information.*
**GWalk-retrieved edited fact:** The chief executive officer of Twitter is Parag Agrawal.
*// This is a correct retrieval because we first identify (in a lossy fashion) entity **twitter** and relation **executive officer** in the KG storing edited facts.*

**MeLLo 2nd Subquestion:** What is the country of citizenship of Elon Musk?
**GWalk 2nd Subquestion:** What is the country of citizenship of Parag Agrawal?
*// MeLLo eventually provides the wrong final answer because the rest of its subquestion is about Elon Musk, though it should be about Parag Agrawal. We note that this is an editing dataset, so the ground truth answers often don't reflect the situation in the real world.*

---

Similarly, here's MeLLo retrieving an unrelated edited fact on an unedited subquestion (MQuAKE-Remastered-CF, case #70).

---

**Unedited Subquestion Example**

**Question:** What is the capital of the country where Premam originated?
**1st Subquestion:** Where Premam was originated?
**Generated Answer (by LLM):** Premam was originated in India.
**MeLLo-retrieved fact:** Carnatic music was created in the country of Poland.
*// Unrelated edited fact retrieved even if this subquestion is not edited.*
**GWalk-retrieved fact:** None.
*// No edited fact is retrieved because no triple (via lossy mapping) on the KG has a source of **Premam** with a relation of **originated in**.*

**MeLLo 2nd Subquestion:** What is the name of the capital city of Poland?
**GWalk 2nd Subquestion:** What is the capital city of India?
*// MeLLo again eventually provides the wrong final answer because the rest of its subquestion is about Poland, though it should be about India.*

---

We also present the quantitative result of the case study between MeLLo (Zhong et al., 2023) and our GWalk. The experiment is conducted using Qwen/Qwen2.5-7B-Instruct (Yang et al., 2024a) on MQUAKE-REMASTERED-CF-3K. We chose QWen2.5-7B-Instruct as the evaluation model primarily because it is the second-best model we evaluated after llama3.1-8b which MeLLo fails to make any edits, according to Table 18.

Figure 3: This is the color legend, where "Correct Case" means the case is answered correctly. "Incorrect Breakdown Path" refers to a case having an incorrect subquestion breakdown path. "Incorrect Edited Retrieval" means the case has an incorrect fact retrieved on an edited subquestion. "Incorrect Unedited Retrieval" means the method retrieves an irrelevant fact where there should be no edited facts. "Others" refers to other issues that occurred evaluating this case, primarily knowledge aggregation. A failing case may contain multiple types of error, but we only count the type that occurs the first.

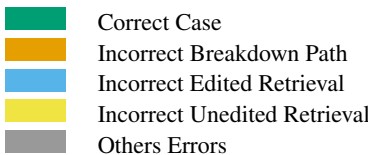

Figure 4: Comparison of MeLLo and GWalk for Editnum of 1

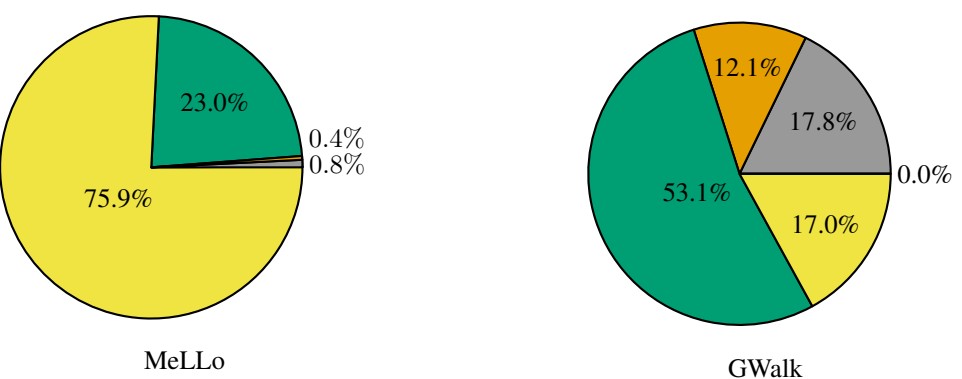

Figure 5: Comparison of MeLLo and GWalk for Editnum 100

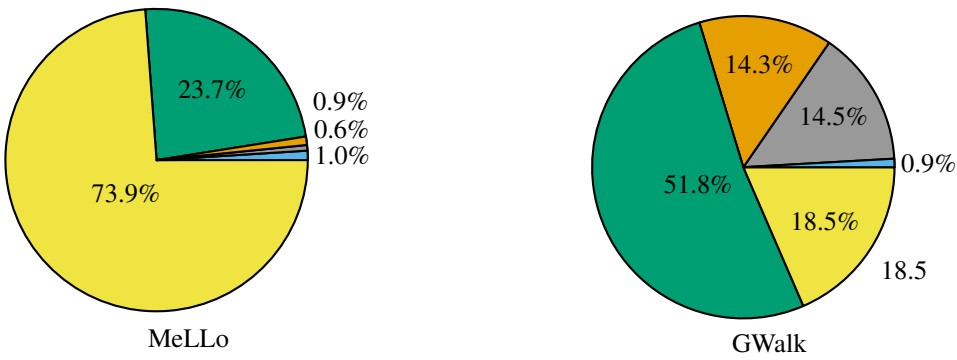

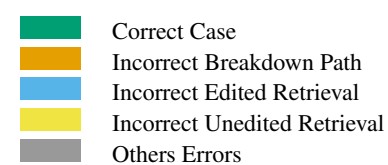

Figure 6: Comparison of MeLLo and GWalk for Editnum 1000

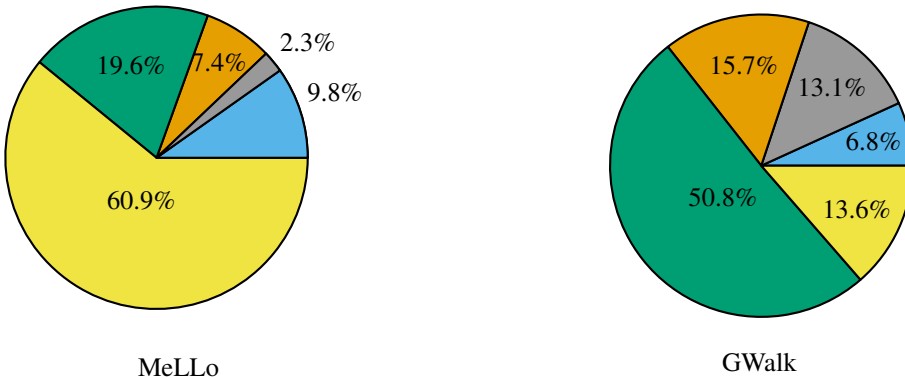

Figure 7: Comparison of MeLLo and GWalk for Editnum 3000

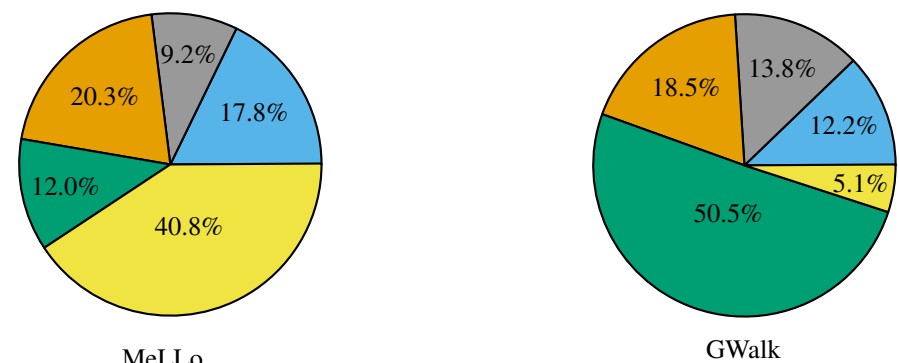

## I ADDITIONAL EXPERIMENT RESULTS

One observation we made in §6.1 is in-context learning-based methods — like ICE (Cohen et al., 2023) and IKE (Zheng et al., 2023a) tend to "OOM" when facing a larger amount of edited facts. This is because these two methods — originally designed for single-edit tasks — essentially dump all edited facts as a long concatenated prompt and expect the model to figure out the corresponding editings naturally. They face OOM issues because when the number of editing facts grows, the prompt becomes extremely long and, therefore, introduces a large amount of KV cache and poses significant memory footprint issues.

While efficiently and effectively handling long input is out-of-scope of our work, as general guidance, we refer interested readers to efficient long context-handing survey/benchmark works like Yuan et al. (2024), which cover the schools and performance of several popular long context-handling methods. Other than the system challenges, another necessary aspect is to improve LLM long context performance, as most LLMs are pre-trained on limited context length and thus cannot effectively handle long input even if the system challenge is addressed. In this regard, we again recommend survey/benchmark works like Lu et al. (2024) for insights. Further, one can certainly convert this long context scenario to leverage the power of the RAG pipeline, much like the majority of multi-hop knowledge editing methods featured in this work.

Table 17: This is the benchmark results of MQUAKE-REMASTERED-T. The reported format is:

*Total Accuracy*
*(Edited Accuracy, Unedited Accuracy)*

| Method | MQUAKE-REMASTERED-T | | | |
|---|---|---|---|---|
| | **1-edit** | **100-edit** | **500-edit** | **1864-edit** |
| lmsys/vicuna-7b-v1.5 (Zheng et al., 2023b) | | | | |
| MeLLo (Zhong et al., 2023) | 19.31 (100, 19.27) | 18.88 (45.0, 17.4) | 22.16 (40.4, 15.47) | 44.37 (44.37, N/A) |
| ICE (Cohen et al., 2023) | <1 | <1 | <1 | OOM |
| IKE (Zheng et al., 2023a) | <1 | <1 | <1 | OOM |
| DeepEdit Wang et al. (2024) | <1 | <1 | <1 | <1 |
| GWalk (Ours) | **35.52** (100, 35.48) | **46.51** (49.0, 46.37) | **48.93** (56.0, 46.33) | **54.88** (54.88, N/A) |
| mistralai/Mistral-7B-Instruct-v0.2 (Jiang et al., 2023) | | | | |
| MeLLo (Zhong et al., 2023) | 10.3 (0, 10.31) | 10.25 (59.0, 7.48) | 18.78 (48.4, 7.92) | 47.75 (47.75, N/A) |
| ICE (Cohen et al., 2023) | <1 | <1 | <1 | OOM |
| IKE (Zheng et al., 2023a) | <1 | <1 | <1 | OOM |
| DeepEdit Wang et al. (2024) | <1 | <1 | <1 | <1 |
| GWalk (Ours) | **34.07** (0, 34.08) | **45.76** (47, 45.69) | **46.78** (51.2, 45.16) | **50.7** (50.7, N/A) |
| meta-llama/Meta-Llama-3-8B-Instruct (AI@Meta, 2024) | | | | |
| MeLLo (Zhong et al., 2023) | <1 | 1.13 (17, <1) | 4.72 (17.4, <1) | 16.58 (16.58, N/A) |
| ICE (Cohen et al., 2023) | <1 | <1 | <1 | OOM |
| IKE (Zheng et al., 2023a) | <1 | <1 | <1 | OOM |
| DeepEdit Wang et al. (2024) | 6.49 (0, 6.49) | 8.48 (36.0, 6.92) | 14.74 (36.20, 6.89) | 34.71 (34.71, N/A) |
| GWalk (Ours) | **70.12** (100, 70.1) | **73.28** (84.0, 72.68) | **76.61** (87, 72.8) | **84.01** (84.01, N/A) |
| meta-Llama/Llama-3.1-8B-Instruct (Dubey et al., 2024) | | | | |
| GWalk (Ours) | **74.68** (100, 74.66) | **76.34** (85, 75.85) | **77.74** (85.4, 74.91) | **83.32** (83.32, N/A) |
| Qwen/Qwen2.5-7B-Instruct (Yang et al., 2024a) | | | | |
| GWalk (Ours) | **44.23** (100, 44.21) | **46.03** (87, 43.71) | **55.1** (85.4, 43.99) | **86.32** (86.32, N/A) |

Table 18: This is the benchmark result for MQuAKE-Remastered-cf-3k reported in the *Total Accuracy* format of: *(Edited Accuracy, Unedited Accuracy)*

| Method | MQuAKE-Remastered-cf-3k | | | |
|---|---|---|---|---|
| | **1-edit** | **100-edit** | **1000-edit** | **3000-edit** |
| lmsys/vicuna-7b-v1.5 (Zheng et al., 2023b) | | | | |
| MeLLo (Zhong et al., 2023) | 16.54 (100, 16.51) | 18 (9.0, 18.31) | 14.63 (8.0, 17.95) | 6.77 (6.77, N/A) |
| ICE (Cohen et al., 2023) | <1 | <1 | OOM | OOM |
| IKE (Zheng et al., 2023a) | <1 | OOM | OOM | OOM |
| DeepEdit (Wang et al., 2024) | <1 | <1 | <1 | <1 |
| GWalk (Ours) | **54.89** (100, 54.87) | **60.9** (54, 61.14) | **57.37** (54.4, 58.85) | **66.33** (66.33, N/A) |
| mistralai/Mistral-7B-Instruct-v0.2 (Jiang et al., 2023) | | | | |
| MeLLo (Zhong et al., 2023) | 19.73 (100, 19.71) | 18.6 (21, 18.52) | 16.33 (17.8, 15.6) | 15.93 (15.93, N/A) |
| ICE (Cohen et al., 2023) | <1 | <1 | OOM | OOM |
| IKE (Zheng et al., 2023a) | <1 | 4.43 (4, 4.49) | OOM | OOM |
| DeepEdit (Wang et al., 2024) | <1 | <1 | <1 | <1 |
| GWalk (Ours) | **56.57** (100, 56.55) | **61.93** (47, 62.45) | **57.17** (51.5, 60.0) | **51.0** (51.0, N/A) |
| meta-llama/Meta-Llama-3-8B-Instruct (AI@Meta, 2024) | | | | |
| MeLLo (Zhong et al., 2023) | <1 | <1 (2.0, <1) | 1.03 (3.0, <1) | 2.3 (2.3, N/A) |
| ICE (Cohen et al., 2023) | <1 | <1 | OOM | OOM |
| IKE (Zheng et al., 2023a) | <1 | <1 | OOM | OOM |
| DeepEdit (Wang et al., 2024) | 22.93 (0, 22.94) | 17.27 (11, 17.48) | 15.03 (15.1, 15.0) | 12.63 (12.63, N/A) |
| GWalk(Ours) | **69.0** (100, 68.99) | **76.73** (67, 77.07) | **75.47** (74.2, 76.1) | **70.6** (70.6, N/A) |
| meta-Llama/Llama-3.1-8B-Instruct (Dubey et al., 2024) | | | | |
| MeLLo (Zhong et al., 2023) | <1 | <1 | <1 | 2.5 (2.5, N/A) |
| DeepEdit (Wang et al., 2024) | 17.80 (100, 17.77) | 15.50 (15, 15.52) | 11.87 (10, 12.8) | 9.47 (9.47, N/A) |
| GWalk (Ours) | **73.3** (100, 73.3) | **76.83** (69, 77.1) | **75.03** (74.6, 75.25) | **71.53** (71.53, N/A) |
| Qwen/Qwen2.5-7B-Instruct (Yang et al., 2024a) | | | | |
| MeLLo (Zhong et al., 2023) | 40.63 (100, 40.61) | 40 (34, 40.21) | 35.23 (23.9, 40.9) | 23.1 (23.1, N/A) |
| DeepEdit (Wang et al., 2024) | 36.43 (100, 36.41) | 29.77 (5, 30.62) | 23.37 (6.3, 31.9) | 5.43 (5.43, N/A) |
| GWalk (Ours) | **65.33** (100, 65.35) | **65.27** (65, 65.28) | **65.07** (68.4, 63.4) | **66.74** (66.74, N/A) |

Table 19: Experiments on MQUAKE-REMASTERED-CF with numbers of edited cases and methods.
*Total Accuracy*
Results are reported in the format: *(Edited Accuracy, Unedited Accuracy)*

| Method | MQUAKE-REMASTERED-CF | | | | |
| --- | --- | --- | --- | --- | --- |
| | **1-edit** | **1000-edit** | **3000-edit** | **6000-edit** | **9171-edit** |
| lmsys/vicuna-7b-v1.5 (Zheng et al., 2023b) | | | | | |
| MeLLo (Zhong et al., 2023) | 22.55 (100, 22.54) | 21.54 (8, 23.2) | 17.79 (7.43, 22.83) | 12.62 (7.28, 22.58) | 6.95 (6.95, N/A) |
| ICE (Cohen et al., 2023) | <1 | OOM | OOM | OOM | OOM |
| IKE (Zheng et al., 2023a) | <1 | OOM | OOM | OOM | OOM |
| DeepEdit Wang et al. (2024) | <1 | <1 | <1 | <1 | <1 |
| GWalk (Ours) | **61.89** (100, 61.89) | **56.98** (56.2, 57.07) | **56.37** (53.97, 57.54) | **54.93** (53.27, 58.06) | **54.15** (54.15, N/A) |
| mistralai/Mistral-7B-Instruct-v0.2 (Jiang et al., 2023) | | | | | |
| MeLLo (Zhong et al., 2023) | 19.83 (<1, 19.84) | 19.08 (20.6, 18.9) | 18.9 (19.47, 18.62) | 18.27 (19.02, 16.87) | 18.09 (18.09, N/A) |
| ICE (Cohen et al., 2023) | <1 | OOM | OOM | OOM | OOM |
| IKE (Zheng et al., 2023a) | <1 | OOM | OOM | OOM | OOM |
| DeepEdit Wang et al. (2024) | <1 | <1 | <1 | <1 | <1 |
| GWalk (Ours) | **61.42** (100, 61.42) | **57.79** (51.8, 58.52) | **56.35** (52.3, 58.32) | **53.73** (50.93, 59.04) | **51.53** (51.53, N/A) |
| meta-llama/Meta-Llama-3-8B-Instruct (AI@Meta, 2024) | | | | | |
| MeLLo (Zhong et al., 2023) | <1 | <1 | <1 | <1 | <1 |
| ICE (Cohen et al., 2023) | <1 | OOM | OOM | OOM | OOM |
| IKE (Zheng et al., 2023a) | <1 | OOM | OOM | OOM | OOM |
| DeepEdit Wang et al. (2024) | 22.16 (100, 22.15) | 19.26 (21.29, 19.01) | 21.09 (24.48, 19.44) | 23.04 (23.77, 21.67) | 24.25 (24.25, N/A) |
| GWalk (Ours) | **74.09** (100, 74.09) | **73.67** (71.1, 73.98) | **72.4** (70.9, 73.13) | **71.62** (70.33, 74.05) | **70.08** (70.08, N/A) |
| meta-Llama/Llama-3.1-8B-Instruct (Dubey et al., 2024) | | | | | |
| GWalk (Ours) | **76.27** (1, 76.27) | **73.48** (73.1, 73.53) | **72.86** (71.98, 73.29) | **72.03** (70.96, 74.08) | **70.94** (70.94, N/A) |
| Qwen/Qwen2.5-7B-Instruct (Yang et al., 2024a) | | | | | |
| GWalk (Ours) | **64.4** (0, 64.41) | **62.61** (66.6, 62.12) | **63.35** (66.34, 61.9) | **64.93** (66.28, 62.4) | **66.79** (66.79, N/A) |

Table 20: Additional experiments on meta-llama/Llama-3.1-8B-Instruct (Dubey et al., 2024) and Qwen/Qwen2.5-7B-Instruct (Yang et al., 2024a) on MQUAKE-REMASTERED-CF-6334. Results are reported in the format: *Total Accuracy* *(Test Edited Accuracy, Train Edited Accuracy, Unedited Accuracy)* .

| Method | MQUAKE-REMASTERED-CF-6334 | | | |
| --- | --- | --- | --- | --- |
| | **100-edit** | **1000-edit** | **3000-edit** | **6344-edit** |
| lmsys/vicuna-7b-v1.5 (Zheng et al., 2023b) | | | | |
| ROME (Meng et al., 2022) | <1 | <1 | <1 | <1 |
| MEND (Mitchell et al., 2022) | 12.75 | 10.36 | 9.56 | 7.24 |
| | (11.11, 11, 13.25) | (7.33, 9.6, 13.64) | (6.1, 7.2, 11.9) | (6.38, 6.49, 10.3) |
| GWalk (Ours) | **57.55** | **61.79** | **59.1** | **56.62** |
| | (22.22, 64.84, 57.48) | (29.08, 66.17, 63.23) | (39.3, 63.74, 64.33) | (44.64, 62.11, 68.25) |
| mistralai/Mistral-7B-Instruct-v0.2 (Jiang et al., 2023) | | | | |
| ROME (Meng et al., 2022) | <1 | <1 | <1 | <1 |
| MEND (Mitchell et al., 2022) | 11.84 | 11.57 | 8.39 | 6.82 |
| | (11.11, 9, 12.36) | (6.95, 8.7, 12.12) | (3.41, 6.6, 10.1) | (2.33, 6.4, 8.4) |
| GWalk (Ours) | **56.25** | **58.9** | **56.03** | **54.43** |
| | (33.33, 57.14, 56.28) | (34.69, 60.57, 60.6) | (42.69, 59.04, 59.85) | (47.49, 57.74, 52.38) |
| meta-llama/Meta-Llama-3-8B-Instruct (AI@Meta, 2024) | | | | |
| ROME (Meng et al., 2022) | <1 | <1 | <1 | <1 |
| MEND (Mitchell et al., 2022) | 13.04 | 13.3 | 9.81 | 7.42 |
| | (11.11, 10, 13.47) | (5.33, 8.4, 14.33) | (4.21, 8.63, 11.1) | (5.12, 7.45, 7.3) |
| GWalk (Ours) | **67.01** | **71.89** | **73.76** | **74.22** |
| | (33.33, 74.73, 66.92) | (47.45, 80.94, 70.65) | (54.05, 81.6, 71.12) | (61.02, 80.47, 73.02) |
| meta-Llama/Llama-3.1-8B-Instruct (Dubey et al., 2024) | | | | |
| GWalk (Ours) | **66.79** | **73.66** | **72.09** | **73.3** |
| | (33.33, 72, 66.66) | (49.47, 73.68, 73.02) | (51.23, 75.1, 70.6) | (55.39, 73.84, 71.55) |
| Qwen/Qwen2.5-7B-Instruct (Yang et al., 2024a) | | | | |
| GWalk (Ours) | **60.59** | **65.42** | **68.75** | **70.49** |
| | (33.33, 62, 60.56) | (30.13, 68.6, 63.83) | (43.65, 69.9, 64.99) | (59.12, 70.51, 68.25) |

