# OpenReview forum: "MQuAKE-Remastered: Multi-Hop Knowledge Editing Can Only Be Advanced with Reliable Evaluations"
_ICLR.cc/2025/Conference — ICLR 2025 Spotlight_

### Official Review · Reviewer_htKb · 2024-11-03

**Soundness:** 3
**Presentation:** 2
**Contribution:** 2
**Rating:** 6
**Confidence:** 4

**Summary:**

This paper identifies significant flaws in the widely used MQuAKE dataset for evaluating multi-hop knowledge editing methods. The authors present MQuAKE-REMASTERED, a corrected and enhanced version of the original dataset. They audit and rectify several errors, including intra-contamination, inner-contamination, and conflicts in the multi-hop question instructions. Furthermore, the authors propose a minimally invasive knowledge editing method called GWalk, which achieves state-of-the-art results without exploiting dataset-specific properties.

**Strengths:**

1. The authors thoroughly analyze the errors in the MQuAKE dataset, bringing awareness to issues that could distort performance evaluations. The audit's depth and transparency add value to the field.
2. MQuAKE-REMASTERED fixes the original dataset's flaws without sacrificing data capacity. This improvement will be a valuable resource for future research on knowledge editing.
3. The proposed GWalk approach is a well-designed and simple method that effectively handles multi-hop knowledge editing without relying on dataset-specific heuristics. Its performance is impressive and highlights the potential for broader applicability.
4. The paper provides extensive re-benchmarking of existing methods, showcasing the significance of the dataset corrections and the robustness of GWalk.

**Weaknesses:**

1. Some methods encounter out-of-memorOOM issues, which restricts broader adoption and testing. The authors could explore efficient data handling or suggest guidelines for reducing memory usage.
2. The experiments cover a few models. Expanding this to a broader range of LLMs (like 5 LLMs) would provide a more generalized understanding of the benchmark's utility.
3. The authors may run controlled experiments to quantify the impact of each type of contamination on model performance. This would provide more insights into which errors had the most significant effect and how much improvement is due to each fix.

**Questions:**

See Weaknesses.

---

> ### Author Response · Authors · 2024-11-19
> **Thanks!**
>
> We thank the reviewer for the detailed review and recognition. We particularly appreciate your suggestion in W3 (method-based impact analysis of different error types) and added this experiment. Here, we address your feedback respectively.
>
> ## **`W1 - Some methods are OOM:` True. However, this is mostly an inevitable design flaw of those methods.**
>
> We believe the reviewer refers to the ICE and IKE methods we featured. These two methods — originally designed for single-edit tasks — essentially dump all edited facts as a long concatenated prompt and expect the model to figure out the corresponding editings naturally.  They face OOM issues because when the number of editing facts grows, the prompt becomes extremely long and introduces a large amount of KV cache.
>
> We briefly discussed this phenomenon around `L479`, **and we appreciate the reviewer's suggestion regarding adding a discussion of efficient handling/guidelines for method design. We now added such a section in Appendix H.** In short, having a long concatenated prompt full of editing facts is effectively a long context-serving scenario. It could benefit from many KV cache compression techniques [1] for reduced memory footprint, and long context extension techniques [2] can help with their performance. Another tested alternative is to do RAG, but this direction kind of reverts back to all the RAG-based methods we featured. We made the connection between such subfields clear in Appendix H.
>
> ---
>
> ## **`W2 - More model coverage (5 LLMs):` Sure, we now added QWen 2.5 and Llama 3.1 for GWalk, DeepEdit and MeLLo, making up a total of 5 LLMs.**
>
> We totally agree with the reviewer that more model coverage can always be better, though we'd like to respectfully note that because of the extensive evaluation we have done per model, expanding the model coverage *fully* would result in a huge computation burden.
>
> For instance, we evaluated 9000+ unedited cases even if there is only 1 edited fact, whereas most prior arts only test this 1 edited case. We are also the only work that evaluated the CF-9k dataset (something the original MQuAKE paper — with an authorship lineup implying decent resource access — didn't even provide results on, citing compute constraints) and done so with much finer editing granularity, as shown in Table 12.
>
> For such reasons, it is unlikely we can provide full coverage (all methods & settings) on two more LLMs during the period of rebuttal. **The best alternative we can do now is to run MeLLo and GWalk on QWen 2.5 and Llama 3.1, and we promise we'll provide full coverage on the rest should our work be camera-ready.** We'd say since GWalk is the only method that performs consistently well (and much better than the others), it should be practically acceptable for future methods to utilize GWalk solely as their baseline.
>
> **`Update: We now also added DeepEdit results on the two new models.`**
>
> > MQuAKE-Remastered-CF-3k  (Table 16)
> | Model | Method | Acc@1-edit | 100-edit | 1000-edit | 3000-edit |
> |-|-|-|-|-|-|
> | QWen2.5-7B-Instruct | MeLLo | 40.63 | 40 | 35.23 | 23.1 |
> | QWen2.5 | DeepEdit | 36.43 | 29.77 | 23.37 | 5.43|
> | QWen2.5 | GWalk (ours) |**65.33**|**65.27**|**65.07**|**66.74**|
> | Llama-3.1-8B-Instruct | MeLLo | <1 | <1 | <1 | 2.5 |
> | Llama-3.1 | DeepEdit |17.8|15.5| 11.87 | 9.47 |
> | Llama-3.1 | GWalk (ours) |**73.3**|**76.83**|**75.03**|**71.53**|
>
> ---
>
> ## **`W3 - Add controlled experiments to quantify the impact of different errors:` Great suggestion, we now adopted this with pleasure.**
>
> This is a great suggestion. In our paper, we mostly focus on quantifying the scale of each type of error by counting its numerical representation within the dataset (like the number of cases), but we never actually quantify their impact to an actual method. We now adopted the reviewer's suggestion and provided the following results.
>
> > MQuAKE-CF-3K with Llama-3.1-8B-Instruct and GWalk (Table 11)
> | Type of Error Fixed | Acc.@100-edit | 1000-edit | 3000-edit |
> |-|-|-|-|
> | None | 45.47 | 42.73 | 39.57 |
> | Inner Contamination | 46.83 | 53.3 | 71.36 |
> | Intra Contamination | 71.17 | 61.73 | 39.87 |
> | Missing Instruction | 49.33 | 47.2 | 45.1 |
> | All (ours Remastered) | **76.83** | **75.03** | **71.51** |
>
> The observed impact is consistent with our analysis: The Inner Contamination fix has the most impact when editing intensity is high (e.g., 3000-edit). Yet, the intra-contamination fix has the most impact with lower editing intensity (e.g., 100-edit). The Missing Instruction fix consistently improves performance across all editing intensities. We provide a more detailed version of this table in Appendix D, Table 11.
>
> (Note, we opt to exclude minor errors like duplicated questions and conflicting edits, as there aren't many of them in this CF-3k dataset).
> ---
> [1] KV Cache Compression, But What Must We Give in Return? A Comprehensive Benchmark of Long Context Capable Approaches
> [2] A Controlled Study on Long Context Extension and Generalization in LLMs

---

> ### Author Response · Authors · 2024-11-28
> **More quantitative result on failure pattern analysis.**
>
> ## **Previously at `W3`, The reviewer called for error impact analysis of a method, which we addressed. Now, we have identified an additional way to collect quantitative results in terms of failure patterns, which paints a related picture from a different angle.**
>
> While the reviewer's `W3`  mainly pertained to the impact of different errors on a method — a great suggestion that we have happly supplied that analysis [above](https://openreview.net/forum?id=m9wG6ai2Xk&noteId=1ZaHrwNh8d) using GWalk as a strong baseline, which highlights the clearer impact of dataset errors. Given the acute, data-oriented observations made by the reviewer, we sense that you might also be interested in an **error pattern analysis**—i.e., how different methods fail even when all dataset errors are fixed. Thus, we present some new quantitative results here.
>
>
> ---
>
> There are roughly four main failure patterns for a multi-hop knowledge editing method, defined as follows:
>
> - **`Incorrect Edited Retrieval`:** A method retrieves an incorrect edited fact on an edited subquestion, as seen in our first case study above.
> - **`Incorrect Unedited Retrieval`:** A method retrieves an irrelevant edited fact where no edited fact should influence the unedited question.
> - **`Incorrect Breakdown Path`:** Errors occur in subquestion breakdown paths — e.g., for a 3-hop question, the method fabricates a 4th hop. This error is only considered after retrieval errors are ruled out, as incorrect retrieval almost always leads to subsequent false breakdowns.
> - **`Other Errors`:** A catch-all category, mainly encompassing various knowledge aggregation failures — i.e., the edited fact is correctly retrieved, but the LLM still relies on its innate (unedited) knowledge.
>
> > With these definitions, we present **MeLLo vs. GWalk failure pattern analysis on MQuAKE-Remastered-CF-3k using `Qwen/Qwen2.5-7B-Instruct`.** This model is chosen for its strong performance with MeLLo, making the comparison meaningful.
> > We note that this analysis is limited to the first instruction (of three total) for each multi-hop question. In full experiments, all three instructions are run, requiring at least one correct answer. This leads to a slight drop in overall accuracy for all methods. **All results are in %.**
> ### **MeLLo Failure Pattern Analysis**
> | # Edits | Correct Case | Incorrect Breakdown Path | Incorrect Edited Retrieval | Incorrect Unedited Retrieval | Other Errors |
> |---------|--------------|-------------------|-----------------------------|------------------------------|--------------|
> | 1-edit  | 23.0         | 0.4                      | 0.0                         | 75.9                         | 0.8          |
> | 100-edit| 23.7         | 0.9                      | 1.0                         | 73.9                         | 0.6          |
> | 1000-edit| 19.6        | 7.4                      | 9.8                         | 60.9                         | 2.3          |
> | 3000-edit| 12.0        | 20.3                     | 17.8                        | 40.8                         | 9.2          |
>
> ### **GWalk Failure Pattern Analysis**
> | # Edits | Correct Case | Incorrect Breakdown Path | Incorrect Edited Retrieval | Incorrect Unedited Retrieval | Other Errors |
> |---------|--------------|--------------------------|-----------------------------|------------------------------|--------------|
> | 1-edit  | 53.1         | 12.1                     | 0.0                         | 17.0                         | 17.8         |
> | 100-edit| 51.8         | 14.3                     | 0.9                         | 18.5                         | 14.5         |
> | 1000-edit| 50.8        | 15.7                     | 6.8                         | 13.6                         | 13.1         |
> | 3000-edit| 50.5        | 18.5                     | 12.2                        | 5.1                          | 13.8         |
>
> ---
>
> ### **Analysis**
> GWalk consistently outperforms MeLLo by retrieving more relevant facts for the edited hop and avoiding irrelevant edits for unedited hops. For `1-edit` where there is literally only one edited fact at play, MeLLo fails **75.9%** of cases due to retrieving this irrelevant edit on many unedited hops and being influenced by it. Such unintended retrieval often leads to hallucinations, incorrectly generating the next subquestion. For example, as shown in our [second case study to reviewer `ND8E`](https://openreview.net/forum?id=m9wG6ai2Xk&noteId=nWjq1OLudd), the irrelevant entity `Poland` was introduced into the reasoning flow, misdirecting it away from `India`.
>
> In contrast, GWalk's topology-based retrieval minimizes unintended retrievals, preserving the reasoning flow and enabling more accurate multi-hop reasoning.
>
> **We hope these additional results clarify your concerns regarding why GWalk is more performant. We also provide pie-chart visualizations of the above tables in [Appendix G, page 23](https://openreview.net/pdf?id=m9wG6ai2Xk#page=23).**

---

> ### Author Response · Authors · 2024-12-03
> **A gentle reminder, and an invitation to discuss.**
>
> We apologize in advance for spamming your inbox, but with the reviewer posting deadline approaching, we would like to gently remind the reviewer to check out our rebuttal and join the discussion, as resolving concerns is something we take very seriously, even with positive ratings.
>
> It might be fair to characterize your main concerns as two-fold: 1) expanding LLM coverage (specifically to 5 models), and 2) conducting error influence analysis with respect to actual performance (not just error count).
>
> While we admit we didn’t address #1 to the fullest capacity (covering all methods, datasets, models, and granularity) due to time and computational limitations, **we did expand the model coverage to 5 LLMs and featured 3 representative methods under a complete set of granularities.** We believe the results are consistent with the core message we aim to convey and will provide the rest during the camera-ready phase. Additionally, we would like to note that none of the prior work on MQuAKE has undergone this thorough of an evaluation — not even the authors of MQuAKE, who presumably have access to greater manpower and resources.
>
> **We found your suggestion regarding #2 to be excellent.** Not only did we fulfill your original request exactly as you asked, we also proactively conducted a failure pattern analysis of GWalk and MeLLo. This can be viewed as a methodological counterpart to your request and provides unique, quantifiable insights that we believe will interest the MHKE community.
>
> We would greatly appreciate it if the reviewer could take a quick look and evaluate whether our current presentation merits an updated rating.

---

### Official Review · Reviewer_WNuN · 2024-11-04

**Soundness:** 3
**Presentation:** 3
**Contribution:** 3
**Rating:** 6
**Confidence:** 4

**Summary:**

MQUAKE has been a primary benchmark for multi-hop knowledge editing, but the authors reveal that up to 33% of MQUAKE’s ground truth labels and 76% of its questions are compromised by clerical or procedural errors. This paper provides a comprehensive audit and correction of these issues, resulting in MQUAKE-REMASTERED, which preserves the dataset’s capacity while enhancing accuracy. Additionally, the authors benchmark various MQUAKE-evaluated editing methods on the revised dataset and note that many approaches overfit the original MQUAKE dataset by exploiting dataset-specific properties. The paper concludes by recommending a minimally invasive approach for faithfully addressing such datasets, demonstrating that it achieves excellent editing results without exploiting dataset idiosyncrasies.

**Strengths:**

1.  Knowledge editing is an important topic, and addressing multi-hop knowledge editing is a challenging yet meaningful direction.
2.  The authors identified and addressed key issues within the MQUAKE dataset, which significantly enhances the benchmark’s credibility.
3.   The proposed GWalk method achieves substantial performance improvement over some baseline methods.

**Weaknesses:**

1.  I believe that methods like fine-tune-based, meta-learning-based, and locate-and-edit would not be significantly affected by the issues raised by the authors in Section 3.1, Section 3.2 and Section 3.3, since they typically perform editing and evaluation on individual samples independently.
2.  GWalk, as a knowledge graph-based search method, is quite common in the Retrieval-Augmented Generation (RAG) domain, which limits its contribution in terms of novelty for knowledge editing.
3.  Although the GWalk method achieves impressive results, there is insufficient analysis on why it performs well. Specifically, the paper lacks a detailed explanation of the benefits introduced by different components, and the construction and search within the knowledge graph could be computationally expensive.
 4.  While the dataset proposed by the authors is helpful for research in the editing domain, especially for multi-hop knowledge editing, its benefits seem to be primarily demonstrated through parameter-preserving methods, such as in-context learning and external knowledge integration. The authors are encouraged to explore more parameter-modifying methods, such as fine-tuning, meta-learning-based, and locate-and-edit approaches, to evaluate the overall effectiveness and versatility of the dataset.

**Questions:**

See weaknesses.

---

> ### Author Response · Authors · 2024-11-19
> **Thanks! (1/2)**
>
> We thank the reviewer for your careful review and recognition. We recognize the main concern of your review (missing non-parameter-preserving editing methods) and provide additional results and discussions. Here, we address your concerns and questions in details.
>
> ---
>
> ## **`W1 & W4 - Would finetuning/Meta-learning/locate-and-edit based methods be less affected by contamination/conflict-natured errors?` We believe they will still be influenced, and more severely than RAG-based methods. But maybe we misunderstood your concerns so please let us know.**
>
> We would say most, if not all, learning-based methods are even more sensitive to contamination/conflict-like errors. Because such type of errors are essentially providing different answers to the same question, so it is like training a model with conflicting labels. **In the case of RAG-based methods, the conflicts do not take effect if they are not retrieved, but parameter-based methods will be under the influence of such conflicting knowledge at all times, and causing a more severe problem.**
>
> However, we sense that the reviewer is obviously an experienced expert in the editing field, so please do let us know if we misunderstood your concerns, and we will surely address them once again in a faithful manner. By the reviewer's wording of *"they typically perform editing and evaluation on individual samples independently"* we suspect the **maybe reviewer means a single-edit setting where there is only 1 edited fact to be addressed at a time.** If so, we discussed this exact setting around `L519`. In short, this setting — by dataset design — is indeed naturally immune to conflicting-like errors, as there isn't another edited fact to conflict with. However, we consider this setting to be an oversimplification of real-world editing scenarios, where systems must handle multiple edits simultaneously, as there sure is more than one edit to address.
>
> ---
>
> Per the findings of prior arts like MeLLo, most general editing methods like ROME and MEND do not perform well under multi-hop scenarios, so we forwent featuring them in our submission. **But this sure is a reasonable request to benchmark them, and here we provide additional results for ROME (locate-then-edit) [1] and MEND (meta-learning) [2]:**
>
> > Vicuna-7B-v1.5
> | Method | Acc@100-edit | 1000-edit | 3000-edit | 6334-edit |
> |-|-|-|-|-|
> ROME | <1 | <1 | <1 | <1 |
> MEND | 12.75 | 10.36 | 9.56 | 7.24 |
> GWalk (ours) | **57.55** | **61.79** | **59.1** | **56.62** |
>
> > Mistral-7B-Instruct-v0.2
> | Method | 100-edit | 1000-edit | 3000-edit | 6334-edit |
> |-|-|-|-|-|
> ROME | <1 | <1 | <1 | <1 |
> MEND | 11.85 | 11.57 | 8.39 | 6.82 |
> GWalk (ours) | **56.25** | **58.9** | **56.03** | **54.43** |
>
> > Meta-Llama-3-8B-Instruct
> | Method | 100-edit | 1000-edit | 3000-edit | 6334-edit |
> |-|-|-|-|-|
> ROME | <1 | <1 | <1 | <1 |
> MEND | 13.04 | 13.3 | 9.81 | 7.42 |
> GWalk (ours) | **67.01** | **71.89** | **73.76** | **74.22** |
>
>
> We'd say the advantage of GWalk is pronounced. We also provide a more detailed report of the above results in Table 19, where we break down the total accuracy into finer subcategories.
>
>
> ---
>
> ## **`W2 - GWalk has limited contribution/novelty because it is RAG-based:` Most performant MHKE methods are RAG-based. However, we do agree that GWalk has limited novelty, which is precisely why we dropped it in a dataset auditing paper instead of making it a standalone method.**
>
> As discussed above, most, if not all, performant multi-hop knowledge editing (MHKE) methods are RAG-based. So we do agree that it is *"quite common,"* and we said exactly this in our paper.
>
> > `L365`: This school of editing methods has proven to be successful, mainly because it can leverage the power of retrieval-argument generation (RAG) combined with the in-context learning (ICL)...
>
> In fact, we won't even shy away from further admitting the core recipe of GWalk —  storing data in a graph format and conducting topology-based retrieval — lacks technical novelty, as this concept is plenty straightforward. We just made the observation about its strong connection in MHKE and executed it nicely (more in our `W3` response below). The lack of technical contribution is precisely why we featured GWalk in a paper mainly focusing on dataset auditing instead of writing a method paper of its own, as we believe the field doesn't need us to sugar-coat something simple into a full-blown paper.
>
> As mentioned in Sec 5.1, we propose the GWalk method for the sole purpose of showing one can achieve good performance without leveraging dataset idiosyncrasies. It stands as a simple yet efficient baseline for all future-coming MHKE methods. **We believe having this kind of baseline is important for a field's advancement since it doesn't make much sense to chase for technical novelty (or sometimes, complexity) if such a proposed solution cannot significantly outperform a simple baseline like GWalk.** And the field of MHKE didn't have a utility baseline like this before GWalk.

---

> ### Author Response · Authors · 2024-11-19
> **Thanks! (2/2)**
>
> (cont.)
>
>
> ## **`W3 - Insufficient analysis on why GWalk performs well:` We now expanded it and added case studies. The core recipe of GWalk is via KG-based retrieval (instead of NL), so there is less chance of unintended retrieval of editing facts.**
>
> We briefly shared our understanding of why GWalk performs well around `L369 - 377`. For your convenience, we provide an abstraction here. We believe GWalk is performant and practical because of two ingredients:
>
> 1. It only stores edited facts in its Editing Knowledge Bank (ref. Figure 2), contrary to some baselines (e.g., RAE), where unedited facts are also stored. This is more practical to maintain as there are always fewer edited facts to keep track of, yet the total search space is much smaller, allowing more precise and efficient retrieval.
>
> 2. Unlike most baselines, which store edited facts in natural language (NL) format (e.g., MeLLo and the majority of existing works) and conduct retrieval based on NL sentence embeddings, we store such editing facts on a Knowledge Graph (KG). The topology-based retrieval greatly reduces unintended retrieval, which almost always causes hallucinations.
>
> Here is a concrete example from MQuAKE-Remastered-CF (case #16), where MeLLo retrieves an incorrect edited fact on an edited subquestion.
>
> > Question: What is the country of citizenship of Twitter's CEO?
> 1st Subquestion: Who is Twitter's CEO?
> Generated answer (by LLM): Twitter's CEO is Elon Musk.
> **MeLLo-retrieved edited fact:** The chief executive officer of CBS Corporation is Steve Jobs. // `Incorrect edited fact retrieved` because this edited fact is close to the subquestion from an embedding standpoint, even if it doesn't provide relevant information.
> **GWalk-retrieved edited fact:** The chief executive officer of Twitter is Parag Agrawal. // This is a correct retrieval because we first identify (in a lossy fashion) entity `twitter` and relation `executive officer` in the KG storing edited facts.
> **MeLLo 2nd subquestion:** What is the country of citizenship of Elon Musk?
> **GWalk 2nd subquestion:** What is the country of citizenship of Parag Agrawal?
> // MeLLo eventually provides the wrong final answer because the rest of its subquestion is about Elon Musk, though it should be about Parag Agrawal. We note that this is an editing dataset, so the ground truth answers often don't reflect the situation in the real world.
>
> Similarly, here's MeLLo retrieving an unrelated edited fact on an unedited subquestion (MQuAKE-Remastered-CF, case #70):
>
> > Question: What is the capital of the country where Premam originated?
> 1st Subquestion: Where Premam was originated?
> Generated answer (by LLM): Premam was originated in India.
> **MeLLo-retrieved fact:** Carnatic music was created in the country of Poland. // `Unrealetd edited fact retrieved` even if this subquestion is not edited.
> **GWalk-retrieved fact:** None // No edited fact is retrieved because no triple (via lossy mapping) on the KG has a source of `Premam` with a relation of `originated in`.
> **MeLLo 2nd subquestion:** What is the name of the capital city of Poland?
> **GWalk 2nd subquestion:**  What is the capital city of India?
> // MeLLo again eventually provides the wrong final answer because the rest of its subquestion is about Poland, though it should be about India.
>
>
> (This is also why MeLLo is less affected by the errors within the dataset, as shown in Table 3, as its retrievals are often wrong to start with.)
>
> Another reviewer (`ND8E`) made a good point of adding case studies, which we think would clearly demonstrate the impact of relevant GWalk ingredients. We have added such cases to Appendix G.
>
> ---
>
> (Btw, *"dataset idiosyncrasies"* is a nice term that conveys the message better than our original choice of "data-specific properties," which sounds like not necessarily a bad thing to leverage. We now stole/borrowed it for our updated manuscript and thank you for the — likely unintentional — suggestion.)
>
> ---
>
> [1] Locating and Editing Factual Associations in GPT
> [2] Fast Model Editing at Scale
> (both methods require finetuning operations)

---

> > ### Comment · Reviewer_WNuN · 2024-11-25
> > **Response to Authors**
> >
> > Thank you for your response. Most of my concerns have been resolved, and I will raise my score.

---

> > > ### Author Response · Authors · 2024-11-26
> > > **Thank you for the score bump, do let us know if there is anything else.**
> > >
> > > We thank the reviewer for being engaging and acknowledging our rebuttal. We are, of course, delighted to see that our response has brought your rating from negative to positive.
> > >
> > > Now that all reviewers have provided positive ratings (`6666`), our first-hand experience told us it is not uncommon for submissions with such ratings to still face rejection. With this in mind, if there is anything further we can do to clarify your concerns — whether or not it warrants an even improved score — please do not hesitate to let us know. We are committed to utilizing the extended discussion period to address any remaining uncertainties or requests you may have.
> > >
> > > Thank you again for your thoughtful feedback and for engaging with our work.
> > >
> > > Sincerely,
> > > *Paper13349* Authors

---

> ### Author Response · Authors · 2024-11-28
> **More quantitative result to W3 — A failure pattern analysis of GWalk and MeLLo.**
>
> ## **`W3 follow-up:` Previously at `W3`, the reviewer called for further analysis on why GWalk performs well, which we addressed with some case studies. Now, we have identified an additional way to collect quantitative results in terms of failure patterns and paints a more comprehensive picture.**
>
> There are roughly four main failure patterns for a multi-hop knowledge editing method, defined as follows:
>
> - **`Incorrect Edited Retrieval`:** A method retrieves an incorrect edited fact on an edited subquestion, as seen in our first case study above.
> - **`Incorrect Unedited Retrieval`:** A method retrieves an irrelevant edited fact where no edited fact should influence the unedited question.
> - **`Incorrect Breakdown Path`:** Errors occur in subquestion breakdown paths — e.g., for a 3-hop question, the method fabricates a 4th hop. This error is only considered after retrieval errors are ruled out, as incorrect retrieval almost always leads to subsequent false breakdowns.
> - **`Other Errors`:** A catch-all category, mainly encompassing various knowledge aggregation failures — i.e., the edited fact is correctly retrieved, but the LLM still relies on its innate (unedited) knowledge.
>
> > With these definitions, we present **MeLLo vs. GWalk failure pattern analysis on MQuAKE-Remastered-CF-3k using `Qwen/Qwen2.5-7B-Instruct`.** This model is chosen for its strong performance with MeLLo, making the comparison meaningful.
> > We note that this analysis is limited to the first instruction (of three total) for each multi-hop question. In full experiments, all three instructions are run, requiring at least one correct answer. This leads to a slight drop in overall accuracy for all methods. **All results are in %.**
> ### **MeLLo Failure Pattern Analysis**
> | # Edits | Correct Case | Incorrect Breakdown Path | Incorrect Edited Retrieval | Incorrect Unedited Retrieval | Other Errors |
> |---------|--------------|--------------------------|-----------------------------|------------------------------|--------------|
> | 1-edit  | 23.0         | 0.4                      | 0.0                         | 75.9                         | 0.8          |
> | 100-edit| 23.7         | 0.9                      | 1.0                         | 73.9                         | 0.6          |
> | 1000-edit| 19.6        | 7.4                      | 9.8                         | 60.9                         | 2.3          |
> | 3000-edit| 12.0        | 20.3                     | 17.8                        | 40.8                         | 9.2          |
>
> ### **GWalk Failure Pattern Analysis**
> | # Edits | Correct Case | Incorrect Breakdown Path | Incorrect Edited Retrieval | Incorrect Unedited Retrieval | Other Errors |
> |---------|--------------|--------------------------|-----------------------------|------------------------------|--------------|
> | 1-edit  | 53.1         | 12.1                     | 0.0                         | 17.0                         | 17.8         |
> | 100-edit| 51.8         | 14.3                     | 0.9                         | 18.5                         | 14.5         |
> | 1000-edit| 50.8        | 15.7                     | 6.8                         | 13.6                         | 13.1         |
> | 3000-edit| 50.5        | 18.5                     | 12.2                        | 5.1                          | 13.8         |
>
> ---
>
> ### **Analysis**
> GWalk consistently outperforms MeLLo by retrieving more relevant facts for the edited hop and avoiding irrelevant edits for unedited hops. For `1-edit` where there is literally only one edited fact at play, MeLLo fails **75.9%** of cases due to retrieving this irrelevant edit on many unedited hops and being influenced by it. Such unintended retrieval often leads to hallucinations, incorrectly generating the next subquestion. For example, as shown in our [second case study](https://openreview.net/forum?id=m9wG6ai2Xk&noteId=nqCx1hrLex), the irrelevant entity `Poland` was introduced into the reasoning flow, misdirecting it away from `India`.
>
> In contrast, GWalk's topology-based retrieval minimizes unintended retrievals, preserving the reasoning flow and enabling more accurate multi-hop reasoning.
>
> **We hope these additional results clarify your concerns regarding why GWalk is more performant. We also provide pie-chart visualizations of the above tables in [Appendix G, page 23](https://openreview.net/pdf?id=m9wG6ai2Xk#page=23).**

---

### Official Review · Reviewer_3g3q · 2024-11-04

**Soundness:** 3
**Presentation:** 3
**Contribution:** 3
**Rating:** 6
**Confidence:** 2

**Summary:**

This paper argues that the original MQuAKE dataset contains data contamination, potentially leading to inaccurate evaluation results. The authors audit several contamination categories and remaster the dataset through rewriting and masking.

**Strengths:**

The paper identifies potential issues within the original MQuAKE dataset and provides an effective remedy. It re-benchmarks several existing knowledge-editing methods and proposes a data-specialty-free approach (GWalk) to address this issue.

**Weaknesses:**

However, the heuristic approach using dynamic masking may not be adaptable to all types of knowledge editing methods, particularly those without memory-based editing.

**Questions:**

- As extensive masking is applied in MQuAKE-Remastered, did the authors evaluate its effectiveness on methods that do not have memory retrieval?

- What instruction was used to prompt Llama-3.1 for rewriting? Was this rewriting model the original version, or was it specifically fine-tuned?

- (Minor) There are a few typos that need correction, such as the duplicate “only” in line 71 and the misspelled “audix” in line 488.

---

> ### Author Response · Authors · 2024-11-19
> **Thanks!**
>
> We thank the reviewer for your observant review and recognition. Here, we address your concerns and questions.
>
> ---
> ## **`W1 & Q1 - The dynamic masking fix is incompatible with non-memory-based methods:` This is true. We actually thought about this and offered CF-6334 as an alternative for such methods. Also here's results on ROME and MEND.**
>
> This is a fair and observant criticism. Dynamic masking essentially adjusts the dataset at test time, making it incompetent for parameter-based methods, which require a constant and consistent set of edited facts across train/test sets at all times.
>
> **Though, in our defense, we did the best we could in this regard:**
>
> * First, the original MQuAKE dataset doesn't even have a train-test split (it is eval-only), so it does not support most learning-required methods to start with. Some literature managed to make those methods run in a "hacky" way (e.g., using the difference between CF-9k and CF-3k as the train set and CF-3k as the test set; more details around L351), but this is a bad practice because the edited facts are not consistent between the two sets.
>
> * We emphasized and briefly discussed the incompatibility between masking-based datasets and parameter-based methods around `L335` and further rolled out a dataset called **MQuAKE-Remastered-CF-6334** (with its details in Appendix D.1). This dataset includes 6334 cases filtered from the CF-9k total set, which are totally error-free. **We also offered multiple official train-test splits (at different editing intensities) on it with a constant and consistent set of edited facts, as parameter-based methods required.**
>
> Per the findings of prior arts like MeLLo, most general editing methods like ROME and MEND do not perform well under multi-hop scenarios, mostly because it is hard to capture the intricate multi-hop relationship without relying on some forms of in-context learning capability, so we forwent featuring them in our first submission. But this sure is a reasonable request, and **here we provide additional results for ROME [1] and MEND [2]:**
>
> > Vicuna-7B-v1.5
> | Method | Acc@100-edit | 1000-edit | 3000-edit | 6334-edit |
> |-|-|-|-|-|
> ROME | <1 | <1 | <1 | <1 |
> MEND | 12.75 | 10.36 | 9.56 | 7.24 |
> GWalk (ours) | 57.55 | 61.79 | 59.1 | 56.62 |
>
> > Mistral-7B-Instruct-v0.2
> | Method | Acc@100-edit | 1000-edit | 3000-edit | 6334-edit |
> |-|-|-|-|-|
> ROME | <1 | <1 | <1 | <1 |
> MEND | 11.85 | 11.57 | 8.39 | 6.82 |
> GWalk (ours) | 56.25 | 58.9 | 56.03 | 54.43 |
>
> > Meta-Llama-3-8B-Instruct
> | Method | Acc@100-edit | 1000-edit | 3000-edit | 6334-edit |
> |-|-|-|-|-|
> ROME | <1 | <1 | <1 | <1 |
> MEND | 13.04 | 13.3 | 9.81 | 7.42 |
> GWalk (ours) | 67.01 | 71.89 | 73.76 | 74.22 |
>
> We'd say the advantage of GWalk is pronounced. We also provide a more detailed report of the above results in Table 19, where we break down the total accuracy into finer subcategories.
>
> ---
>
> ## **`Q2 - Rewriting details w.r.t Llama 3.1:` We now share the rewriting prompt in Appendix E.2. And yes, it is the original `meta-llama/Llama-3.1-405B` model.**
>
> We use the original `meta-llama/Llama-3.1-405B` with few-shot demonstrations to rewrite the instructions.
>
> > Instruction:
> > Given a chain of relations, generate 3 multi-hop questions that comprehensively include the semantics of the relations.
> >
> > Example 1:
> > Relation Chain:
> > XXX -> 'The author of {{}} is' -> '{{}} is a citizen of' -> ?
> >
> > Generated Questions:
> > 1. What is the country of citizenship of the author of XXX?
> > 2. What country is the author of XXX a citizen of?
> > 3. What is the nationality of the author of XXX?
> >
> > Example 2:
> > Relation Chain:
> > XXX -> '{{}} was developed by' -> 'The chairperson of {{}} is' -> '{{}} is a citizen of' -> '{{}} is located in the continent of' -> ?
> >
> > Generated Questions:
> > 1. What continent is the country located in, where the chairperson of the developer of XXX is a citizen?
> > 2. On which continent is the country located, whose citizen is the chairperson of the company that developed XXX?
> > 3. Which continent houses the country of the chairperson of the developer of XXX?
> >
> > Example 3:
> > ...
> >
> > Relation Chain:
> > <The relational chain we want the generated questions to be based on>
>
> We now add this prompt to Appendix E.2 of our updated manuscript.
>
> ---
> ## **`Q3 - Typos:` Thanks! We have now fixed the ones you mentioned and several others we caught.**
>
> And we will definitely give it another polish should our work reach camrea-ready.
>
> ---
>
> [1] Locating and Editing Factual Associations in GPT
> [2] Fast Model Editing at Scale

---

> > ### Comment · Reviewer_3g3q · 2024-11-24
> >
> > Thank you for providing detailed responses. Most of my comments are addressed.

---

> > > ### Author Response · Authors · 2024-11-25
> > > **Thank you!**
> > >
> > > If the reviewer feels there is anything else we could better address, please do not hesitate to let us know. On the same note, if the reviewer does not believe our work warrants a higher rating than `6`, we hope our clarifications can potentially improve your confidence in your assessment.
> > >
> > > We’d also like to point out that your **W1** (compatibility with non-memory-based methods) largely overlaps with the main concerns raised by reviewer `WNuN`, who is similarly interested in such methods and is the only reviewer to give a negative rating of (`5`). **As reviewers ourselves, we want to refrain from overloading your inbox, so we would like to take this opportunity to, of course, thank you, and also invite reviewers with similar concerns to join the discussion.**
> > >
> > > We believe we have addressed these concerns faithfully and directly by highlighting the rationale behind our CF-6334 dataset, which was specifically designed to support learning-required methods at the time of submission. Furthermore, we provide direct results for ROME (locate-then-edit) and MEND (meta-learning) as requested. These results demonstrate the significant advantages of GWalk and confirm the field’s intuition regarding the strengths of learning-based methods on the MHKE task.

---

### Official Review · Reviewer_ND8E · 2024-11-09

**Soundness:** 3
**Presentation:** 3
**Contribution:** 3
**Rating:** 6
**Confidence:** 3

**Summary:**

This paper introduces MQuAKE-Remastered, an enhanced version of the MQuAKE dataset designed to improve the evaluation of multi-hop knowledge editing methods for large language models (LLMs). The authors identify significant flaws in the original MQuAKE dataset, including data contamination, conflicting edits, missing information in multi-hop question instructions, and duplicated cases. These issues, which affect lots of the data including 33% or 76% of MQUAKE’s questions and ground truth labels, undermine the accuracy of previous evaluations conducted with MQuAKE. The revised dataset, MQuAKE-Remastered, provides a cleaner, more reliable benchmark without sacrificing dataset capacity. Additionally, the authors propose a novel, minimally invasive method for knowledge editing, achieving state-of-the-art performance without exploiting dataset-specific biases. The paper provides re-benchmarks of existing methods on MQuAKE-Remastered, offering a clearer view of each method's true effectiveness and guiding future approaches to knowledge editing evaluation.

**Strengths:**

1.	The proposed fixed benchmark and new baseline are clear and easy to follow.
2.	The analysis for the prior benchmark and re-benchmarking the prior works are extremely valuable, which prevents the community from going in the wrong direction of research.
3.	The proposed fixing method is useful for building reliable fixed benchmark.
4.	The proposed method is an effective brand new baseline for this task.

**Weaknesses:**

1.	The process of identifying the error cases is unclear. The paper introduces the types of error cases, but not show the process of extracting them.

**Questions:**

1.	How to identify the problematic cases of the original MQuAKE Benchmark?
2.	Do you have a more detailed analysis and case study of your GWalk methods? Could you show some correct cases in your fixed benchmark where GWalk performs well while the baselines fail? What’s the critical part of the GWalk method?

---

> ### Author Response · Authors · 2024-11-19
> **Thank you! (1/2)**
>
> We thank the reviewer for your detailed review and recognition. *"Preventing the community from going in the wrong direction of research"* is exactly our work's main motivation, and we are glad that message is well-received. Here, we address your concerns and questions.
>
> ---
> ## **`W1 & Q1 - The process of error detection is unclear`: Fair criticism. We have a section delivering a similar message (post-audit correctness checking); we now form it for both and expand its details.**
>
>
> This is a fair criticism. We roughly shared how we detected those errors in Appendix C, though we formed it under the context of (post) *Audit Correctness Analysis* in our initial submission. This is largely the same information the reviewer requested (error detection process), as our post-analysis is basically rerunning our error detection procedure to ensure there are no apparent misses.
>
> **We now updated Appendix C's title to entail both subjects (error detection procedure & post-audit double checking) and gave more details in the updated PDF.** For your convenience, here's a high-level summary:
>
> * Intra/Inner Contamination: We iterate each subquestion of a case respectively. We then mark a subquestion as contaminated if there is an edit having the same subject and relation as an unedited subquestion.
> * Conflicting Edits: We check if there are edit-reflecting triples that share the same subject and relation, but with different tails/targets. We replace the minority edit with the majority edit and update the subsequent answers.
> * Missing Info in Instructions: We defined a list of keywords for each relation and identified cases where instructions lacked any corresponding keywords for each of their relations. Using LLaMA 3.1-405B, we regenerated the instructions with few-shot demonstrations. We rerun the check and manually inspect/fix the small number of flagged cases post-fixing.
> * Duplicated Cases: We check if there are any cases sharing 100% identical pre and post-editing paths and remove the duplications.

---

> ### Author Response · Authors · 2024-11-19
> **Thank you (2/2)**
>
> (posting a new comment because with our two case studies we went about the character limit of one reply.)
>
> ## **`Q2 - Case studies of GWalk over Baselines / What's its core recipe?` Sure, here are some cases. The core recipe of GWalk is via KG-based retrieval (instead of NL), so there is less chance of unintended retrieval of editing facts.**
>
>
> We briefly shared our understanding of why GWalk performs well around `L369 - 377`. For your convenience, we provide an abstraction here. We believe GWalk is performant and practical because of two ingredients:
>
> 1. It only stores edited facts in its Editing Knowledge Bank (ref. Figure 2), contrary to some baselines (e.g., RAE), where unedited facts are also stored. This is more practical to maintain as there are always fewer edited facts to keep track of, yet the total search space is much smaller, allowing more precise and efficient retrieval.
>
> 2. Unlike most baselines, which store edited facts in natural language (NL) format (e.g., MeLLo and the majority of existing works) and conduct retrieval based on NL sentence embeddings, we store such editing facts on a Knowledge Graph (KG). The topology-based retrieval greatly reduces unintended retrieval, which almost always causes hallucinations.
>
> Here is a concrete example from MQuAKE-Remastered-CF (case #16), where MeLLo retrieves an incorrect edited fact on an edited subquestion.
>
> > Question: What is the country of citizenship of Twitter's CEO?
> 1st Subquestion: Who is Twitter's CEO?
> Generated answer (by LLM): Twitter's CEO is Elon Musk.
> **MeLLo-retrieved edited fact:** The chief executive officer of CBS Corporation is Steve Jobs. // `Incorrect edited fact retrieved` because this edited fact is close to the subquestion from an embedding standpoint, even if it doesn't provide relevant information.
> **GWalk-retrieved edited fact:** The chief executive officer of Twitter is Parag Agrawal. // This is a correct retrieval because we first identify (in a lossy fashion) entity `twitter` and relation `executive officer` in the KG storing edited facts.
> **MeLLo 2nd subquestion:** What is the country of citizenship of Elon Musk?
> **GWalk 2nd subquestion:** What is the country of citizenship of Parag Agrawal?
> // MeLLo eventually provides the wrong final answer because the rest of its subquestion is about Elon Musk, though it should be about Parag Agrawal. We note that this is an editing dataset, so the ground truth answers often don't reflect the situation in the real world.
>
> Similarly, here's MeLLo retrieving an unrelated edited fact on an unedited subquestion (MQuAKE-Remastered-CF, case #70):
>
> > Question: What is the capital of the country where Premam originated?
> 1st Subquestion: Where Premam was originated?
> Generated answer (by LLM): Premam was originated in India.
> **MeLLo-retrieved fact:** Carnatic music was created in the country of Poland. // `Unrealetd edited fact retrieved` even if this subquestion is not edited.
> **GWalk-retrieved fact:** None // No edited fact is retrieved because no triple (via lossy mapping) on the KG has a source of `Premam` with a relation of `originated in`.
> **MeLLo 2nd subquestion:** What is the name of the capital city of Poland?
> **GWalk 2nd subquestion:**  What is the capital city of India?
> // MeLLo again eventually provides the wrong final answer because the rest of its subquestion is about Poland, though it should be about India.
>
>
> (This is also why MeLLo is less affected by the errors within the dataset, as shown in Table 3, as its retrievals are often wrong to start with.)
>
>
> **The reviewer made a good point of adding case studies, which we think would clearly demonstrate the impact of relevant GWalk ingredients.** We have added such cases to Appendix G.

---

> ### Author Response · Authors · 2024-11-28
> **More quantitative result to Q2 — A failure pattern analysis of GWalk and MeLLo.**
>
> ## **`Q2 follow-up - Case studies of GWalk over Baselines?` The reviewer called for case studies, we did that, and now we figure out an additional way to collect some quantitative results in terms of failure patterns, which paints a more comprehensive picture.**
>
> There are roughly four main failure patterns for a multi-hop knowledge editing method, defined as follows:
>
> - **`Incorrect Edited Retrieval`:** A method retrieves an incorrect edited fact on an edited subquestion, as seen in our first case study above.
> - **`Incorrect Unedited Retrieval`:** A method retrieves an irrelevant edited fact where no edited fact should influence the unedited question.
> - **`Incorrect Breakdown Path`:** Errors occur in subquestion breakdown paths — e.g., for a 3-hop question, the method fabricates a 4th hop. This error is only considered after retrieval errors are ruled out, as incorrect retrieval almost always leads to subsequent false breakdowns.
> - **`Other Errors`:** A catch-all category, mainly encompassing various knowledge aggregation failures — i.e., the edited fact is correctly retrieved, but the LLM still relies on its innate (unedited) knowledge.
>
> > With these definitions, we present **MeLLo vs. GWalk failure pattern analysis on MQuAKE-Remastered-CF-3k using `Qwen/Qwen2.5-7B-Instruct`.** This model is chosen for its strong performance with MeLLo, making the comparison meaningful.
> > We note that this analysis is limited to the first instruction (of three total) for each multi-hop question. In full experiments, all three instructions are run, requiring at least one correct answer. This leads to a slight drop in overall accuracy for all methods. **All results are in %.**
> ### **MeLLo Failure Pattern Analysis**
> | # Edits | Correct Case | Incorrect Breakdown Path | Incorrect Edited Retrieval | Incorrect Unedited Retrieval | Other Errors |
> |---------|--------------|--------------------------|-----------------------------|------------------------------|--------------|
> | 1-edit  | 23.0         | 0.4                      | 0.0                         | 75.9                         | 0.8          |
> | 100-edit| 23.7         | 0.9                      | 1.0                         | 73.9                         | 0.6          |
> | 1000-edit| 19.6        | 7.4                      | 9.8                         | 60.9                         | 2.3          |
> | 3000-edit| 12.0        | 20.3                     | 17.8                        | 40.8                         | 9.2          |
>
> ### **GWalk Failure Pattern Analysis**
> | # Edits | Correct Case | Incorrect Breakdown Path | Incorrect Edited Retrieval | Incorrect Unedited Retrieval | Other Errors |
> |---------|--------------|--------------------------|-----------------------------|------------------------------|--------------|
> | 1-edit  | 53.1         | 12.1                     | 0.0                         | 17.0                         | 17.8         |
> | 100-edit| 51.8         | 14.3                     | 0.9                         | 18.5                         | 14.5         |
> | 1000-edit| 50.8        | 15.7                     | 6.8                         | 13.6                         | 13.1         |
> | 3000-edit| 50.5        | 18.5                     | 12.2                        | 5.1                          | 13.8         |
>
> ---
>
> ### **Analysis**
> GWalk consistently outperforms MeLLo by retrieving more relevant facts for the edited hop and avoiding irrelevant edits for unedited hops. For `1-edit` where there is literally only one edited fact at play, MeLLo fails **75.9%** of cases due to retrieving this irrelevant edit on many unedited hops and being influenced by it. Such unintended retrieval often leads to hallucinations, incorrectly generating the next subquestion. For example, as shown in our [second case study](https://openreview.net/forum?id=m9wG6ai2Xk&noteId=nWjq1OLudd), the irrelevant entity `Poland` was introduced into the reasoning flow, misdirecting it away from `India`.
>
> In contrast, GWalk's topology-based retrieval minimizes unintended retrievals, preserving the reasoning flow and enabling more accurate multi-hop reasoning.
>
> **We hope these additional results clarify your concerns regarding why GWalk is more performant. We also provide pie-chart visualizations of the above tables in [Appendix G, page 23](https://openreview.net/pdf?id=m9wG6ai2Xk#page=23).**

---

> > ### Author Response · Authors · 2024-12-03
> > **A gentle reminder, and an invitation to discuss.**
> >
> > As the reviewer posting deadline approaches, we would like to gently remind the reviewer to check out our rebuttal and join the discussion, as resolving concerns is something we take very seriously, even with positive ratings.
> >
> > Your two main concerns are 1) the need for more clarity regarding the error detection procedure in our audit and 2) further analysis on why GWalk works. To address these, we have [restructured and expanded the post-audit correctness checking section](https://openreview.net/forum?id=m9wG6ai2Xk&noteId=2lS8bGU1Iy) to address the former. For the latter, we have included a [case study](https://openreview.net/forum?id=m9wG6ai2Xk&noteId=6oPNbBdXCU) (as requested) and conducted a [quantitative failure pattern analysis](https://openreview.net/forum?id=m9wG6ai2Xk&noteId=6oPNbBdXCU) (going a step beyond the reviewer's original request, if we may say so ourselves).
> >
> > We would greatly appreciate it if the reviewer could take a quick look and evaluate whether our current presentation merits an updated rating.

---

### Author Response · Authors · 2024-12-04
**Summary of Reviewers' Feedback and Our Rebuttals**

We are grateful to receive an all-positive review of `6666`. Historically, this rating has had a relatively decent chance of acceptance; that said, we authors are true believers in the philosophy that "ratings are not everything" and we genuinely appreciate the discretionary attention from the AC — for better or worse — therefore, we provide a summary of our work, the feedback we received, and our rebuttals for the convenience of the AC and the general public.

---

## **TL;DR of Our Work**

Our work provides a thorough audit of the original MQuAKE datasets [1] — a popular and ONLY available set of reflective multi-hop knowledge editing datasets, often serving as the SOLE evaluation benchmark for many proposed methods. Through our audit, we revealed that **up to 33% to 76% of MQuAKE’s questions and ground truth labels are corrupted**.

To address these issues, we introduced a fix called *MQuAKE-Remastered*, which corrects the errors across all MQuAKE datasets without sacrificing dataset capacity. Furthermore, we re-benchmarked almost all open-sourced multi-hop knowledge editing methods specifically evaluated on MQuAKE using our Remastered dataset. Our findings revealed that many strong existing methods exploit dataset idiosyncrasies unique to MQuAKE. In response, we proposed a simple yet highly effective solution, *GWalk*, which achieves strong performance without relying on non-generalizable operations.

So in short, our contributions are four-fold: we **audited** MQuAKE, we **fixed** its errors, we **re-benchmarks** almost all available methods, and we offered **a simple baseline solution that is way beyond SOTA** (up to 53.84% improvement) while utilizing a faithful approach.

---

## **Reviewers' Recognition**

Given the nature of our submission, we believe much of the impact of our work lies in the **following three criteria**:

1. Whether the topic (errors in MQuAKE) we are addressing is important and well-motivated.
2. Whether our solution (auditing MQuAKE and fixing it to Remastered) is proper and thorough.
3. Whether our re-benchmark is contributive and comprehensive.

We are glad to report reviewers find our work to score the trifecta of the above merits, as:


* **Reviewers find the topic we are addressing to be important and well-motivated.**
    * `ND8E` *"The analysis... prevents the community from going in the wrong direction of research."* *"The proposed ... is useful for building reliable fixed benchmark.*"
    * `WNuN` *"Knowledge editing is an important topic, and addressing multi-hop knowledge editing is a challenging yet meaningful direction."*
    * `htKb` *"This improvement will be a valuable resource for future research on knowledge editing."*


* **Reviewers find our audits and fixes to MQuAKE to be effective and comprehensive.**
    *  `ND8E` *"The revised dataset ... provides a cleaner, more reliable benchmark without sacrificing dataset capacity."*
    *  `3g3q` *"The paper identifies potential issues ... and provides an effective remedy."*
    * `WNuN` *"This paper provides a comprehensive audit and correction of ..."* *"The authors identified and addressed ... which significantly enhances the benchmark’s credibility."*
    * `htKb` *"The authors thoroughly analyze the errors in the MQuAKE dataset, bringing awareness ... The audit's depth and transparency add value to the field."*



* **Reviewers appreciate our re-benchmark of existing methods on the error-fixed datasets.**
    * `ND8E` *"The paper provides re-benchmarks of ... offering a clearer view of each method's true effectiveness and guiding future approaches to knowledge editing evaluation."*
    *  `WNuN` *"the authors benchmark various MQUAKE-evaluated editing methods on the revised dataset and note that many approaches overfit the original MQUAKE dataset by exploiting dataset-specific properties."*
    *  `htKb` *"The paper provides extensive re-benchmarking of existing methods, showcasing the significance of the dataset corrections and the robustness of GWalk."*


* **Additionally, many reviewers also appreciate our pilot baseline method — GWalk — for being performant yet without leverging dataset idiosyncrasies.**
    * `ND8E` *"the authors propose a novel, minimally invasive method for knowledge editing, achieving state-of-the-art performance without exploiting dataset-specific biases"* *"The proposed method is an effective brand new baseline for this task."*
    *  `WNuN` *"The proposed GWalk method achieves substantial performance improvement over some baseline methods."*  *"...demonstrating that it achieves excellent editing results without exploiting dataset idiosyncrasies."*
    *  `htKb` *"The proposed GWalk approach is a well-designed and simple method that effectively handles multi-hop knowledge editing without relying on dataset-specific heuristics. Its performance is impressive and highlights..."*


**We appreciate the recognition and couldn't ask for more for this type of work.**

---

> ### Author Response · Authors · 2024-12-04
> **Reviewers' Concerns and Our Rebuttals.**
>
> A summary is only fair if it also faithfully highlights the concerns raised by our reviewers, so here we are. Other than some content/context/motivation clarifications and presentation suggestions, the reviewers are concerned about our work in the following ways:
>
>
> * **`ND8E` requests more clarity on our error-identification procedure — added.**
>     * *"...The paper introduces the types of error cases, but not show the process of extracting them."*
>     * We have a section delivering a similar message (post-audit correctness checking). We now form it for both (identification & post-checking) and expand its details. The reviewer has yet to engage our rebuttal, but we believe we have precisely delivered what was asked.
>
> * **`3g3q` and `WNuN` worry about the compatibility of our Remastered datasets with parameter-based/learning-required methods, with `WNuN` requesting evaluation results on such methods — we have a CF-6334 dataset made specifically for such methods, and we now added the requested evaluation results.**
>     * *"the heuristic approach using dynamic masking may not be adaptable to all types of knowledge editing methods...."*
>     * *"The authors are encouraged to explore more parameter-modifying methods, such as ... to evaluate the overall effectiveness and versatility of the dataset."*
>     * We have explained the Remastered datasets requiring dynamic masking indeed are not compatible with learning-required methods. However, this is an inherited issue of MQuAKE as it is presented to be eval-only (thus does not really support learning-based methods to start with), and we have done our best in making a CF-6334 dataset out of the CF-9k that is suitable for such methods. We also benchmarked ROME (locate-then-edit) and MEND (meta-learning) on such dataset.
>     * **Both reviewers have acknowledged our rebuttal.**
>
> * **`htKb` wants us to expand our coverage to 5 LLMs — done, at least partially.**
>     * *"Expanding this to a broader range of LLMs (like 5 LLMs) would provide a more generalized understanding.."*
>     * Due to time and resource constraints, we cannot feature two more models to the fullest capacity (covering all methods, datasets, models, and granularity) within the rebuttal window, but we did expand the model coverage to 5 LLMs and featured 3 representative methods under a complete set of granularities. The results are consistent with the core message we aim to convey and will provide the rest during the camera-ready phase. Additionally, we would like to note that none of the prior work on MQuAKE has undergone this thorough of an evaluation (see comparison in [Table 10](https://openreview.net/pdf?id=m9wG6ai2Xk#page=10)) — not even the authors of MQuAKE, who presumably have access to greater manpower and resources.
>     * The reviewer has yet to engage our rebuttal. But we believe it is clear that we have gone above and beyond in terms of evaluation coverage — no other work comes close to ours, again per [Table 10](https://openreview.net/pdf?id=m9wG6ai2Xk#page=10).
>
> * **`ND8E` and `WNuN` request more analysis on why GWalk is so performant — we expanded the section discussing this, added case studies as requested, and proactively provided a failure pattern analysis of GWalk and MeLLo as an addition.**
>     * *"Do you have a more detailed analysis and case study of your GWalk methods? Could you show some correct cases in your fixed benchmark where..."*
>     * *"Although the GWalk method achieves impressive results, there is insufficient analysis on why it performs well...."*
>     * Again, we believe we have addressed this issue in a head-on fashion from as many angles as possible. **`WNuN` acknowledged our rebuttal**, but `ND8E` has yet to engage in our discussion.
>
> * **`htKb` wants us to add an error impact analysis w.r.t. methods — done.**
>     * *"The authors may run controlled experiments to quantify the impact of each type of contamination on model performance..."*
>     * We think this is an excellent suggestion and fulfilled it as demanded. Our original impact analysis focused more on the quantitative case count of each type of error, but not on the end-to-end influence on methods — which is clearly something interesting to see. This request also prompted us to do the failure pattern analysis mentioned above, which is the method end of this controlled study. The reviewer has yet to engage, but we have delivered exactly what was asked, and more.
>
> ---
>
> **While it is unfortunate that we did not get to engage with all reviewers, we understand that everyone has priorities. We find many of our reviewers' feedback fair, helpful, and constructive, and we believe we have faithfully addressed all raised concerns.**
>
> ---
>
> [1] MQuAKE: Assessing Knowledge Editing in Language Models via Multi-Hop Questions. EMNLP 2023.
> (It might also be worth noting that we have privately confirmed our findings with the first authors of MQuAKE, and we thank them for being engaging and supportive of community scrutiny.)

---

### Meta-Review · Area_Chair_Q4m1 · 2024-12-20

**Metareview:**

Large Language Models (LLMs) can produce factually incorrect answers due to outdated knowledge or undesirable training outcomes. Knowledge editing techniques aim to patch these inaccuracies efficiently without disrupting other model behaviors. However, real-world knowledge is deeply intertwined, making the propagation of edits across related facts (multi-hop knowledge editing, or MHKE) a complex challenge.

MQuAKE, the most popular dataset for evaluating MHKE methods, has served as a key benchmark. Despite its importance, the authors discovered significant flaws in MQuAKE, compromising its reliability.

Key Contributions

1. Audit of MQuAKE Dataset
   - Found that 33% to 76% of MQuAKE’s questions and labels were corrupted due to clerical and procedural errors.
   - Categorized errors in detail, including mislabeling and ambiguous questions.

2. MQuAKE-Remastered
   - Delivered a comprehensive fix, correcting errors without reducing dataset capacity.
   - Introduced a new version of the dataset, named MQuAKE-Remastered, ensuring better reliability for future evaluations.

3. Re-Benchmarking Knowledge Editing Methods
   - Reassessed all major knowledge editing methods previously evaluated on MQuAKE using the remastered dataset.
   - Exposed that many methods overfit idiosyncrasies specific to the original MQuAKE dataset.

4. Introduction of GWalk
   - Proposed a simple yet highly effective baseline method, GWalk, for MHKE.
   - GWalk achieved up to 53.84% performance improvement over existing methods by avoiding reliance on dataset-specific properties.

Findings

- The flawed nature of the original MQuAKE dataset misrepresented the performance of many knowledge editing methods.
- Dynamic masking techniques used in GWalk enabled more generalizable editing.
- A thorough error analysis and correction significantly improved dataset utility and credibility.

Technical Insights

1. Audit and Fixes
   - Errors included logical inconsistencies, vague questions, and misaligned answers.
   - Auditing involved manual and automated validation to ensure corrections upheld dataset integrity.

2. GWalk Method
   - Minimal invasiveness ensured compatibility across diverse editing scenarios.
   - Avoided exploitation of unique dataset quirks, unlike previous methods.

3. Broader Evaluation
   - Benchmarked methods on five LLMs to generalize findings.
   - Compared methods like ROME and MEND under various evaluation settings.


Conclusion

The paper delivers a foundational improvement to MHKE research by auditing, fixing, and remastering the MQuAKE dataset, creating a reliable benchmark for evaluating knowledge editing methods. GWalk, as a baseline solution, exemplifies how minimally invasive methods can achieve state-of-the-art performance.

This work establishes a robust foundation for advancing multi-hop knowledge editing and highlights the importance of dataset integrity in driving meaningful research.

**Additional Comments On Reviewer Discussion:**

Reviewer Recognition

- Importance: Addressed a critical problem in dataset reliability for MHKE research.
- Effectiveness: Provided comprehensive fixes to MQuAKE, enhancing its benchmark reliability.
- Contribution: Re-benchmarking clarified method performance and offered a robust new baseline (GWalk).

Reviewer Concerns and Rebuttals

1. Error Identification Process
   - Expanded explanations of error identification and post-correction validation.

2. Compatibility with Parameter-Based Methods
   - Introduced a new dataset subset (CF-6334) for parameter-based evaluation.

3. Model Coverage
   - Increased LLM coverage to five, demonstrating consistency across models.

4. Performance Analysis of GWalk
   - Provided case studies and failure pattern analysis to explain GWalk’s superior performance.

5. Error Impact Analysis
   - Added insights on how errors in MQuAKE influenced specific methods.

---

### Decision · Program_Chairs · 2025-01-22

Accept (Spotlight)